# Endophyte genomes support greater metabolic gene cluster diversity compared with non-endophytes in *Trichoderma*

**Kelsey Scott**[1]*, **Zachary Konkel**[1,2], **Emile Gluck-Thaler**[3], **Guillermo E. Valero David**[1], **Coralie Farinas Simmt**[1], **Django Grootmyers**[4], **Priscila Chaverri**[5,6], **Jason Slot**[1,7]

1 Department of Plant Pathology, The Ohio State University, Columbus, OH, United States of America, 2 Center for Applied Plant Sciences, The Ohio State University, Columbus, OH, United States of America, 3 Laboratory of Evolutionary Genetics, University of Neuchâtel, Neuchâtel, Switzerland, 4 Department of Ecology and Evolutionary Biology, University of Tennessee, Knoxville, TN, United States of America, 5 Department of Natural Sciences, Bowie State University, Bowie, MD, United States of America, 6 School of Biology and Natural Products Research Center (CIPRONA), University of Costa Rica, San José, Costa Rica, 7 Center for Psychedelic Drug Research and Education, The Ohio State University, Columbus, OH, United States of America

* scott.1907@osu.edu

**Data Availability Statement:** All relevant data files for this study are publicly available from the Dryad

## Abstract

*Trichoderma* is a cosmopolitan genus with diverse lifestyles and nutritional modes, including mycotrophy, saprophytism, and endophytism. Previous research has reported greater metabolic gene repertoires in endophytic fungal species compared to closely-related non-endophytes. However, the extent of this ecological trend and its underlying mechanisms are unclear. Some endophytic fungi may also be mycotrophs and have one or more mycoparasitism mechanisms. Mycotrophic endophytes are prominent in certain genera like *Trichoderma*, therefore, the mechanisms that enable these fungi to colonize both living plants and fungi may be the result of expanded metabolic gene repertoires. Our objective was to determine what, if any, genomic features are overrepresented in endophytic fungi genomes in order to undercover the genomic underpinning of the fungal endophytic lifestyle. Here we compared metabolic gene cluster and mycoparasitism gene diversity across a dataset of thirty-eight *Trichoderma* genomes representing the full breadth of environmental *Trichoderma*'s diverse lifestyles and nutritional modes. We generated four new *Trichoderma endophyticum* genomes to improve the sampling of endophytic isolates from this genus. As predicted, endophytic *Trichoderma* genomes contained, on average, more total biosynthetic and degradative gene clusters than non-endophytic isolates, suggesting that the ability to create/modify a diversity of metabolites potential is beneficial or necessary to the endophytic fungi. Still, once the phylogenetic signal was taken in consideration, no particular class of metabolic gene cluster was independently associated with the *Trichoderma* endophytic lifestyle. Several mycoparasitism genes, but no chitinase genes, were associated with endophytic *Trichoderma* genomes. Most genomic differences between *Trichoderma* lifestyles and nutritional modes are difficult to disentangle from phylogenetic divergences among species, suggesting that *Trichoderma* genomes maybe particularly well-equipped for lifestyle

database (https://doi.org/10.5061/dryad.
r2280gbhn).

**Funding:** This work was supported by grants from
the National Science Foundation (https://www.nsf.
gov/): (DEB-1638999/DEB-1638976) to JCS and
PC and (DEB-925672/DEB-1019972) to PC. This
work was also supported by a Marie Skłodowska-
Curie Individual Fellowship (https://marie-
sklodowska-curie-actions.ec.europa.eu/) (SEP-
210615880-TRADEOFF) to EGT. The funders had
no role in study design, data collection and
analysis, decision to publish, or preparation of the
manuscript.

**Competing interests:** The authors have declared
that no competing interests exist.

plasticity. We also consider the role of endophytism in diversifying secondary metabolism
after identifying the horizontal transfer of the ergot alkaloid gene cluster to *Trichoderma*.

## Introduction

*Trichoderma* (Hypocreales, Sordariomycetes, Ascomycota) is a cosmopolitan genus of fungi
found growing on many different substrates which define their "lifestyle." The majority of *Trichoderma* species are soil inhabitants. However, many *Trichoderma* species live as endophytes,
i.e., growing asymptomatically beneath the epidermis of a living plant host [1–3]. Some *Trichoderma* species prefer decaying plant matter or other living fungi as a substrate [3, 4]. A subset of
*Trichoderma* species appears to be limited to a single lifestyle/substrate, while other species have
been recorded as having multiple lifestyle strategies [3–6]. Within these lifestyles, *Trichoderma*
species usually exhibit one of two different nutritional modes: mycotrophy/mycoparasitism
(consumption of other fungi) or saprotrophy (consumption of decaying plant matter) [4].

Multiple *Trichoderma* species, including endophytic species, are known for their ability to
produce a variety of secondary metabolites and/or exhibit one or more mycoparasitic mechanisms [1, 7–15]. Some *Trichoderma* secondary metabolites mimic phytohormones, which
impact host plant vigor and trigger plant-wide defenses [16–21]. *Trichoderma* species also produce secondary metabolites that can antagonize invading microorganisms to the benefit of
themselves and their host plants [1, 22, 23]. Furthermore, endophytism is expected to select for
fungal metabolic pathways that provide resistance to defensive metabolites, such as flavonoids,
terpenes, and phenolics, which plants produce constitutively or in response to damage or invasion [24–26]. Specifically, the ability to degrade or modify diverse phenolics might be favored
in endophytic *Trichoderma* species [27–31]. The mechanisms that enable these fungi to colonize both living plants and fungi may be the result of expanded metabolic gene repertoires.
Therefore, we expect that endophytic *Trichoderma* species would have an expanded metabolic
repertoire compared to non-endophytic species.

Fungal metabolic pathways are often encoded in metabolic gene clusters, loci with closely-
linked genes for enzymes, transporters, and regulation of a common pathway [32–36]. Metabolic gene clusters can encode the production of secondary metabolites (biosynthetic gene
cluster, BGC) or their degradation (degradative gene clusters, DGC), among other functions.
The clustering of metabolic gene clusters is evidence of selection on the metabolic function
they perform. Therefore, it can suggest more precise hypotheses about the genetic basis of species ecology than genes in isolation [37, 38]. Fungal BGCs can encode the production of ecologically significant metabolites, such as the antibiotics trichodermin and gliotoxin [39, 40],
and anti-herbivory ergot alkaloids produced by Clavicipitaceae species [27]. By contrast,
DGCs may provide fitness benefits to plant-associated fungi by degrading exogenous toxins,
such as the fungicide cyanate [41] or plant defense compounds, such as benzoxazolinones and
stilbenes [42, 43].

Mycoparasitic fungi produce chitinases, serine proteases, glucanases, other cell-wall degrading enzymes, and antibiotics/antifungals, all of which have been associated with mycoparasitism [44–46]. Chitinase genes are necessary not only for fungal growth and development but
also for the parasitism of competing organisms, such as insects, nematodes, and other fungi
[47, 48]. Similar to the ecological patterns seen in certain gene clusters, chitinase genes have
been identified as being expanded in mycoparasitic fungal lineages and reduced in non-mycoparasitic lineages, likely due to adaptive natural selection [49, 50]. In *Trichoderma*, chitinase

genes in the BI/BII subgroup were reported to be expanded in mycoparasitic species, exhibiting positive selection [51]. Similar to BGCs and DGCs, an expanded repertoire of chitinases and other mycoparasitism-related genes may provide fitness benefits to plant-related fungi by facilitating competition with organisms competing with the fungal endophyte or the endophyte's host plant.

The diversity of genes and gene clusters is reportedly associated with particular fungal lifestyles [26, 50]. For example, endophytic Xylariales species contain a greater diversity of metabolic gene clusters when compared to closely-related non-endophytic species [26]. Similarly, chitinase genes in *Trichoderma* are more diverse in mycoparasites than in saprophytes [50]. Many fungal species are not constrained to a single lifestyle and instead have the capability to interact with a variety of other organisms and substrates, a trait which is potentially explained by their expanded chitinolytic and metabolic diversity [52–54]. Specific fungal gene clusters are reported to be enriched in particular ecological niches [38] possibly due to horizontal gene transfer (HGT) among species with overlapping lifestyles and thus shared selective pressures [30, 35, 55, 56]. Differential metabolic gene cluster diversification [57–59] and loss among fungi in different ecological niches may also contribute to uneven cluster distributions and diversity [32, 35, 52].

In this study, we asked whether endophytic *Trichoderma* genomes have a greater diversity of BGCs and DGCs than non-endophytic *Trichoderma* genomes. We also sought to determine whether the genomes of endophytic *Trichoderma* have more diverse repertoires of chitinase and other mycoparasitism-related genes compared to non-endophytic *Trichoderma* genomes. To address these questions, we sequenced the genomes of four Peruvian isolates of a recently described endophytic species *T. endophyticum*, which is placed in the *T. harzianum* species complex and found as both an endophyte of several species of Neotropical trees [3] as well as living freely in soil [6]. These new genomes were included in a robust comparative database with 34 reannotated publicly available *Trichoderma* genomes that exhibited various lifestyles and nutritional modes to identify ecological patterns. We specifically investigated trends in the composition of BGCs, DGCs, chitinase genes, and other mycoparasitism-associated genes across the 38 genomes.

We found that endophytic *Trichoderma* species have a significantly greater overall diversity of both DGC and BGC than non-endophytic *Trichoderma* species. Still, we found limited evidence that any particular gene cluster families or classes contributed to this overall trend. We did not find a significant association between chitinase diversity and the endophytic lifestyle, though we identified several other mycoparasitism genes that are expanded in endophytic *Trichoderma* genomes. Despite identifying several overarching trends in the genomic features present in endophytic *Trichoderma* species, the interrelatedness between *Trichoderma* lifestyle and phylogeny makes it difficult to infer specific genomic features that could underpin the endophytic lifestyle. Additionally, we identified the horizontal transfer of the ergot alkaloid gene cluster to *Trichoderma*.

## Results

### Four new *T. endophyticum* genomes supplement the *Trichoderma* dataset with high-quality endophytic genomes

Four endophytic isolates of *T. endophyticum* were previously isolated from rubber trees (*Hevea* spp.) in Peru [3]. Isolates PP24 and PP89 were obtained from the trunk of wild *Hevea guianensis* and isolates LA10 and LA29 were obtained from the trunk of wild *Hevea brasiliensis*. All four isolates were sequenced with both Illumina and Nanopore sequencing technology and *de novo* assembled for this study. Twenty-nine additional *Trichoderma* assemblies were

**Table 1. Genome statistics of each of the hybrid-assembly *T. endophyticum* genomes.**

| Strain | BUSCO* (%) | Illumina data (Gbp) | MinION data (Gbp) | Average Illumina Coverage | Number genes** | Contig N50 | Genome length (Mb) |
|---|---|---|---|---|---|---|---|
| LA10 | 99.0 | 3.02 | 0.27 | 77.0x | 12,619 | 1,095,838 | 39.2 |
| LA29 | 99.0 | 3.18 | 0.32 | 81.1x | 12,635 | 1,266,890 | 39.2 |
| PP24 | 99.0 | 2.16 | 1.09 | 55.5x | 12,608 | 713,317 | 38.9 |
| PP89 | 99.1 | 3.09 | 1.10 | 79.3x | 12,606 | 1,414,947 | 38.9 |

*Percent of single copy and complete BUSCO genes present from the Sordariomycetes BUSCO dataset (sordariomyceta_odb9).

**number of genes shown is post-OrthoFiller step.

downloaded directly from NCBI, and six genomes were assembled using publicly available SRA data (S1 Table). The four *T. endophyticum* genomes each have a genome completeness of ≥99.0%, referencing the Sordariomycetes BUSCO dataset (sordariomyceta_odb9) (Table 1). The remaining *Trichoderma* genomes exhibit completeness ranging from 61.2–99.0% (S2 Table). Due to a severely fragmented and incomplete genome (single and complete BUS-COs = 61.2%), *T.* cf. *atroviride* isolate LU140 was included in the phylogenomic tree (S1 Fig) but excluded from all subsequent analyses, resulting in a total of 38 high quality genomes in the final dataset. The four *T. endophyticum* genomes ranged in length between 38.9–39.2Mb (average 39.0 Mb), and the overall full *Trichoderma* genomes in the dataset ranged from 31.7–45.8 Mb (average 37.1 Mb). The *T. endophyticum* genomes contained between 12,606 and 12,635 genes, and the remaining genomes in the dataset ranged from 9,242 and 14,297 genes (Table 1 and S2 Table). The *Trichoderma* clade with the lowest average number of genes per genome was Longibrachiatum (average of 9776 genes per genome), and the clade with the highest was Virens (average of 13,231 genes per genome). The *T. endophyticum* assemblies are available from NCBI under the following BioProject Accession: PRJNA899549.

## Phylogenomic analysis of thirty-eight *Trichoderma* isolates from diverse lifestyles and nutritional modes yields a tree concordant with previous multi-gene phylogenies

We reconstructed a phylogenomic tree of 154 well-supported single-copy orthologs (SCO) (average ≥75% bootstrap percent support [BP] at each node) (S1 Fig). This phylogeny resolves into five distinct clades (100% BP), which are in agreement with previously defined groups in *Trichoderma* [3, 4, 60, 61]. The resolved clades are identified as Harzianum, Virens, Longibra-chiatum, section *Trichoderma*, and Brevicompactum. Harzianum contains *T. endophyticum* (4 isolates), *T. harzianum* (2), *T. simmonsii* (1), *T. guizhouense* (1), an unspecified *Trichoderma* species (OTPB3; 1 isolate), *T. afroharzianum* (2), and *T. plueroticola* (1). Virens contains three *T. virens* isolates. Longibrachiatum consists of *T. reesei* (5), *T. parareesei* (1), *T. longibrachia-tum* (2), *T. bissettii* (1), and *T. citrinoviride* (1). Section *Trichoderma* contains *T. atroviride* (4), *T.* cf. *atroviride* (2), *T. gamsii* (2), *T. koningiopsis* (1), *T. asperellum* (2), and *T. hamatum* (1). Brevicompactum contains *T. brevicompactum* (1) and *T. arundinaceum* (1).

Harzianum contains the majority of the endophytic isolates in this dataset and is composed entirely of isolates with a mycotrophic nutritional mode. All four *T. endophyticum* genomes are located in a single subclade that is well-supported (100% BP). Furthermore, these genomes resolved into separate groups within the *T. endophyticum* species subclade based on their geo-graphical origin/host tree species (PP vs. LA). The *Trichoderma* isolate with an undescribed species (OPTB3), previously considered to be *T. harzianum*, is most closely related to *T. guiz-houense* (NJAU4742). The clade Virens is most closely related to Harzianum, and these clades are well-defined (100% BP).

Virens contains only the three *T. virens* isolates, which all have a mycotrophic nutritional mode and have been reported to live on other fungi, in the soil, and in wood and/or leaf litter (S1 Table).

Longibrachiatum is an entirely saprotrophic clade that lacks any fungi-dwelling or endophytic isolates and is the only clade that does not contain species with more than one reported lifestyle (S1 Table). This clade is well-supported (100% BP) and mostly composed of the five monophyletic (100% BP) *T. reesei* isolates. *Trichoderma bissettii* (JCM 1883) and the two *T. longibrachiatum* genomes (ATC 18648 and SMF2) make up the sister clade to the *T. reesei* genomes (100% BP), and *T. citrinoviride* (TUCIM 6016) diverged earliest in this Longibrachiatum clade.

Section *Trichoderma* is the only clade that contains both saprotrophic and mycotrophic species. All species in this clade can be found living freely in the soil, although most species are also reported to live as either endophytes or in wood/leaf litter. Most of this clade is composed of a subclade (100% BP) containing the four *T. atroviride* and two *T.* cf. *atroviride* isolates, all of which are saprotrophic and can live in both the soil and in wood/leaf litter. The two *T.* cf. *atroviride* isolates (LU140 and LU132) are clustered within this subclade with high support values (100% BP). *Trichoderma koningiopsis* (POS7) is located adjacent to the two *T. gamsii* genomes (A5MH and T6085). Another subclade that is marginally supported in section *Trichoderma* (81% BP) contains the saprotrophic *T. hamatum* (GD12) and two *T. asperellum* genomes (CBS 433.97 and B05). Most endophytic species in this dataset are reported to have a mycoparasitic lifestyle, while *T. hamatum* is one of the two endophytic *Trichoderma* species with a saprotrophic lifestyle.

Brevicompactum is a well-supported clade (100% BP) adjacent to section *Trichoderma* and contains only saprotrophic species. Both species that compose this clade are capable of living freely in the soil, although only *T. brevicompactum* is also found as an endophyte (S1 Table). Additionally, *T. brevicompactum* is the second of the two species in our dataset that is both saprotrophic and endophytic.

## Endophytic isolates have a greater number of DGCs and BGCs compared with non-endophytic isolates

*Trichoderma* genomes contain multiple DGCs, which we have grouped into evolutionarily-related cluster families and gene cluster classes defined by which common "core" processing functions are present in each cluster. We identified DGCs containing salicylate hydroxylase (SAH) (116 total cluster families identified in the dataset), quinate 5-dehydrogenase (QDH) (38 total), aromatic ring-opening dioxygenase (ARD) (8 total), naringenin 3-dioxygenase (NAD) (39 total), ferulic acid decarboxylase (FAD) (41 total), phenol 2-monooxygenase (PMO) (56 total), catechol dioxygenase (CCH) (26 total), benzoate 4-monooxygenase (BPH) (two total), pterocarpan hydroxylase (PAH) (two total). Additionally, we found multiple "hybrid" clusters that contain several different core genes (e.g., hybrid CCH-PMO cluster). On average, a *Trichoderma* genome contains 9.8 DGCs (range 3–16 clusters). One or more gene clusters containing SAH, QDH, PMO, or hybrid of CCH-SAH are present in most *Trichoderma* genomes (Fig 1). PAH clusters are only found in each of the two *T. asperellum* genomes, with a single DGC per genome. BPH clusters are found only in *T. harzianum* (T6776) and *T. gamsii* (T6085) genomes. SAH-SDO clusters are present only in *T. arundinaceum* (IBT-40837) and *T. brevicompactum* (IBT-40841). And a single FAD-SAH cluster was identified in *T. hamatum* (GD12). Despite varying assembly quality throughout the dataset, there is no correlation between the number of identified DGCs and assembly N50 (S2A Fig). The number of identified BGCs per genome positively correlates with the number of identified DGCs per

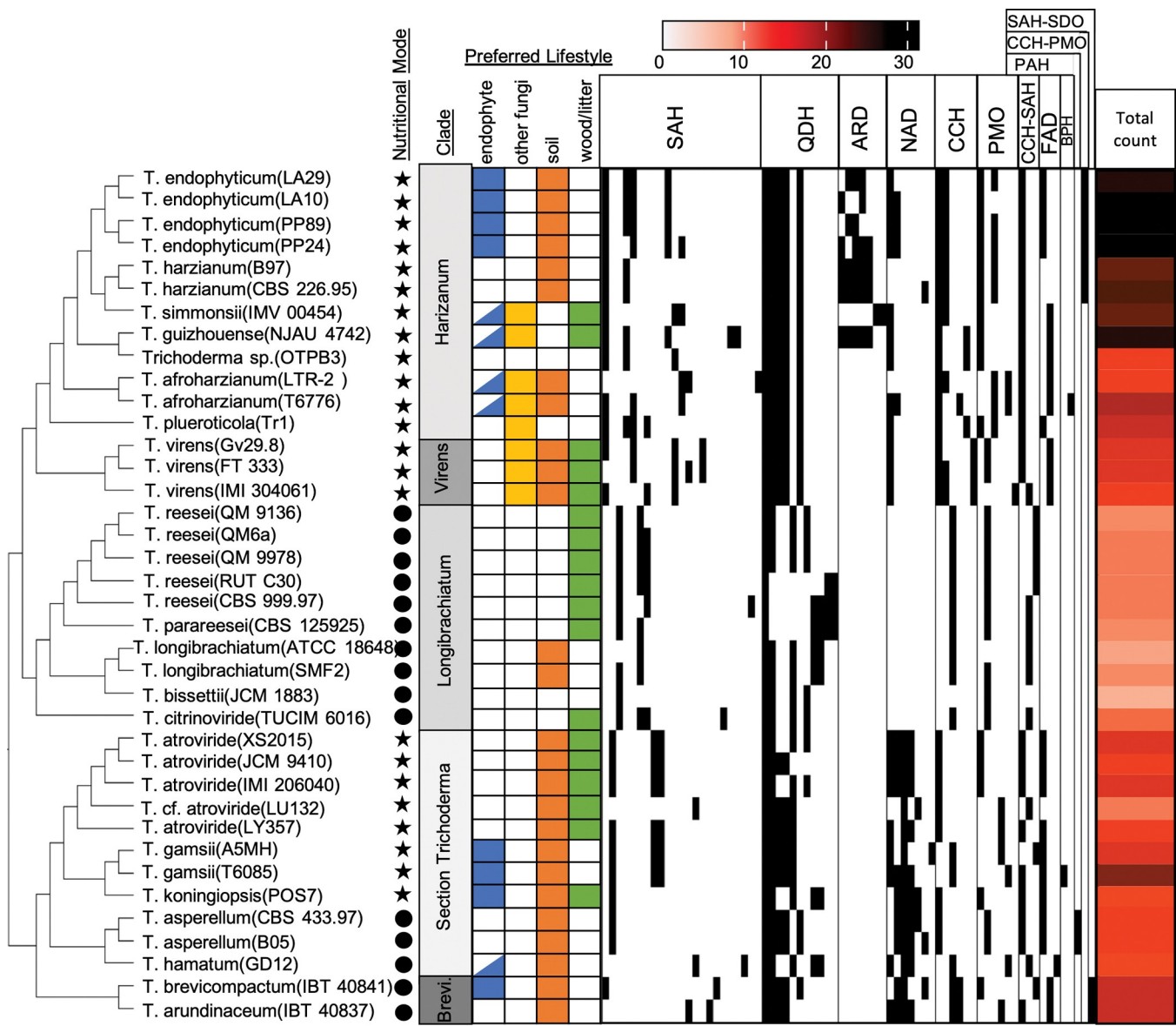

**Fig 1. Distribution and total amount of DGC families per genome varies across isolates and *Trichoderma* clade.** The presence of different DGC families in a particular genome are indicated by a black cell, absence of a particular family is indicated by a white cell. DGC families are separated by cluster class. *Trichoderma* species lifestyle is indicated by presence/absence of color-coordinated cell, nutritional mode is indicated by either a circle (saprotroph) or a star (mycotroph). *Trichoderma* species lifestyle is indicated by the presence or absence of a color-coordinated cell, a potential endophyte lifestyle is indicated by a half-filled cell. Nutritional mode is indicated by either a circle (saprotroph) or a star (mycotroph). The total amount of DGC families identified in each genome is indicated by the red heatmap. Clade "Brevicompactum" is abbreviated as "Brevi".

genome (S3A Fig). The proportion of BGC to the total number of genes within a genome also positively correlates with the proportion of DGC to the total number of genes (S3B Fig). Larger *Trichoderma* genomes have a greater diversity of DGCs (S4A Fig) as well as higher proportions of DGCs to total number of genes (S5A Fig). *Trichoderma* genomes that contain greater numbers of genes also contain greater numbers of DGCs (S6 Fig).

 *Trichoderma* endophytes have a significantly greater number of total DGCs compared to non-endophytes (S7A Fig). Additionally, we compared the number of DGCs within each DGC class (defined by the cluster core gene) between endophyte genomes and non-endophyte

genomes. Endophytic genomes contained greater numbers of SAH, ARD, PMO, and FAD DGCs (S8 Fig).

Results from Blomberg's K [62] and Pagel's Lambda [63, 64] analyses suggest that the individual DGC class counts per genome as well as the overall DGC count per genome is consistent with the *Trichoderma* phylogeny (Blomberg's K and Pagel's Lambda: $p \leq 0.05$) (S9A, S9B Fig). According to Blomberg's K analysis, the DGC classes CCH-PMO and ARD are overdispersed in the *Trichoderma* phylogeny ($K < 0.8$) as well as being overrepresented in endophytic genomes (average count in endophyte genome:average count in non-endophyte genomes [E: NE] $> 2$). However, their counts and distributions are also consistent with Brownian motion evolution (Blomberg's K $p < 0.05$). All other DGC classes, as well as the total number of DGCs per genome, are either not enriched in endophyte genomes (E:NE $< 2$) and/or are consistent with Brownian motion model evolution (Blomberg's K/Pagel's Lambda: $p \leq 0.05$) (S9A, S9B Fig).

Similarly, the calculation of Fritz and Purvis' D-statistic for each individual DGC cluster family suggests that the distribution of most DGC families are consistent with Brownian motion evolution (D-statistic $< 1$, $p > 0.05$) (S10 Fig). Although FAD family 250 and ARD family 215 were found enriched in endophytes (E:NE $> 2$) and were found to be a DGC families enriched in a particular lineage ($D < 0$), the Fritz and Purvis' analysis suggests that the distribution of this DGC family is consistent with the *Trichoderma* phylogeny ($p > 0.05$). All other DGC families either are not enriched in endophytic genomes, have a distribution consistent with Brownian motion, or are not overdispersed in the *Trichoderma* phylogeny.

The distribution of DGC cluster families varies greatly among the *Trichoderma* dataset and primarily reflects the *Trichoderma* phylogeny (Fig 1). Longibrachiatum has the fewest overall number of identified DGCs (S11A Fig), as well as one of the lowest proportions of DGCs to total gene number (S11C Fig). All Longibrachiatum genomes also lack FAD or NAD DGCs, despite these cluster classes being present in the other four clades (Fig 1). Brevicompactum and Harzianum have the highest number of DGCs and proportion of DGCs to total gene number (S11A, S11C Fig). Compared to Harzianum and Virens, there is greater NAD diversity within Brevicompactum and section *Trichoderma* (Fig 1). However, there are no NAD DGCs in two genomes in Harzianum, despite all other Harzianum genomes containing at least one NAD DGC.

The *T. endophyticum* genomes are among the genomes with the highest diversity of DGCs (Fig 1) and have a greater diversity of DGCs clusters than expected, given the association between total gene number and diversity of DGCs (S10 Fig). Each *T. endophyticum* genome contains at least one ARD DGCs and a CCH-PMO DGC, both exclusively identified in Harzianum. In each *T. endophyticum* genome, there is a least one NAD DGC also found in most other genomes, except for genomes in Longibrachiatum. The four *T. endophyticum* genomes have almost identical DGC repertoires, with minimal differences between their overall number of DGCs and individual DGC families between the *T. endophyticum* genomes (Fig 1).

Every *Trichoderma* genome contains multiple BGCs, which we have grouped into families (homologous clusters across species), and further categorized into classes based on the "core" metabolic genes present in each cluster, as assigned by antiSMASH: nonribosomal peptide synthetase (NRPS) (384 total clusters identified in the dataset), type I polyketide synthase (T1PKS) (496 total), NRPS-like (309 total), terpene (304 total), T1PKS-NRPS hybrid clusters (240 total), fungal-associated ribosomally synthesized and posttranslationally modified peptides (fungal-RiPPs) (nine total), beta-lactone (12 total), and other "hybrid" clusters containing two or more of these key genes (51 total). BGCs containing fungal-RiPPs and beta-lactone are found in exclusively in Harzianum genomes (Fig 2, S12 Fig). Additionally, a single indole BGC was identified in the *T. brevicompactum* (IBT40841) genome (not shown in Fig 2). Similar to

the diversity of DGCs in the dataset, there are correlations between different genome features and the number of BGCs in each *Trichoderma* genome. There is no correlation between genome N50 and BGCs identified per genome, or metabolic genes identified per genome (S2B, S2C Fig). Larger *Trichoderma* genomes tend to have greater diversities of BGCs and metabolic genes (S4B, S4C Fig), as well as a greater proportion of BGCs to total genes (S5B Fig). Additionally, genomes with greater numbers of genes tended to have a greater diversity of BGCs (S6B Fig).

Similar to the pattern found in *Trichoderma* DGCs, *Trichoderma* endophyte genomes have a significantly greater number of BGCs when compared to non-endophyte genomes (S7B Fig). Endophytic genomes were found to have significantly greater numbers of T1PKS, NRPS-like, beta-lactone, and fungal-RiPP BGCs (S13 Fig). Phylogenetic correction and assessment of the BGC class counts with Blomberg's K and Pagel's Lambda suggest that the counts of each separate BGC classes are consistent with the Brownian motion model of trait evolution (Blomberg's K and Pagel's Lambda > 0.8, $p \leq 0.05$) (S9A, S9B Fig). Beta-lactone BGCs are found in greater abundance in endophytic genomes (E:NE > 2) and appear to be overdispersed in the *Trichoderma* genus (K < 0.8; however, the distribution of the beta-lactone is ultimately consistent with Brownian model trait evolution (Blomberg's K: $p \leq 0.05$). All other BGC classes we evaluated, as well as the total count of BGCs per genome, despite being overdispersed in the phylogeny (Blomberg's K/Pagel's Lambda < 0.08), are not overrepresented in endophytic genomes (E:NE < 2) and have a distribution consistent with the *Trichoderma* phylogeny (Blomberg's K/Pagel's Lambda: $p \leq 0.05$) (S9A, S9B Fig).

Similarly, the calculation of Fritz and Purvis' D-statistic for each individual BGC family suggests that the distribution of each BGC family can also be explained by the *Trichoderma* phylogeny (D-statistic < 1, $p > 0.05$) (S14 Fig). Terpene family 1276 has a very high D-statistic (D > 4.5), indicating it is overdispersed in *Trichoderma*; however, it is not found overrepresented in endophytic genomes (E:NE < 2), and its distribution was consistent with Brownian motion ($p > 0.05$). Terpene family 1276 is found in all *Trichoderma* genomes besides *T. simmonsii* (IMV 00454). All other BGC families either are not enriched in endophytic genomes (E:NE < 2), have a distribution consistent with Brownian motion (D-statistic: $p > 0.05$), or were not overdispersed in the *Trichoderma* phylogeny (D-statistic < 1).

The distribution of BGC cluster families varies greatly across the *Trichoderma* dataset and primarily reflect the *Trichoderma* phylogeny (Fig 2). There are significant differences in the total diversity of BGCs identified across the different clades (S11B Fig), as well as the proportion of BGCs to total gene number (S11D Fig). Longibrachiatum genomes contain the lowest diversity of BGCs, while Harzianum, Brevicompactum, and Virens contain the highest diversity of BGCs (S11B Fig). The same pattern of BGC diversity is also observed in the proportion of BGCs to total gene number; Longibrachiatum genomes have the lowest proportions, while Harzianum, Brevicompactum, and Virens have the highest proportions of BGCs to total identified genes (S11D Fig).

The *T. endophyticum* genomes have some of the highest diversities of BGCs (Fig 2) and have a greater diversity of BGC clusters than expected, given the association between BCG diversity and genome length (S4B Fig) and total gene number (S6B Fig). The *T. endophyticum* genomes contain a similar number of NRPS (12–13), T1PKS (16–20), NRPS-like (8–9), T1PKS-NRPS (9–10), fungal RiPP (1), and beta-lactone (1) DGCs. All *T. endophyticum* genomes contain a terpene-NRPS (family 1020) hybrid cluster, which is also present in all other Harzianum genomes as well as *T. virens* (IMI 304061). Additionally, *T. endophyticum* LA29 contains an NRPS-indole (family 1016) hybrid cluster, only found in LA29 and *T. virens* (IMV 00454). Overall, the *T. endophyticum* genomes are very similar and have few differences between their total number of BGCs and individual BGC families.

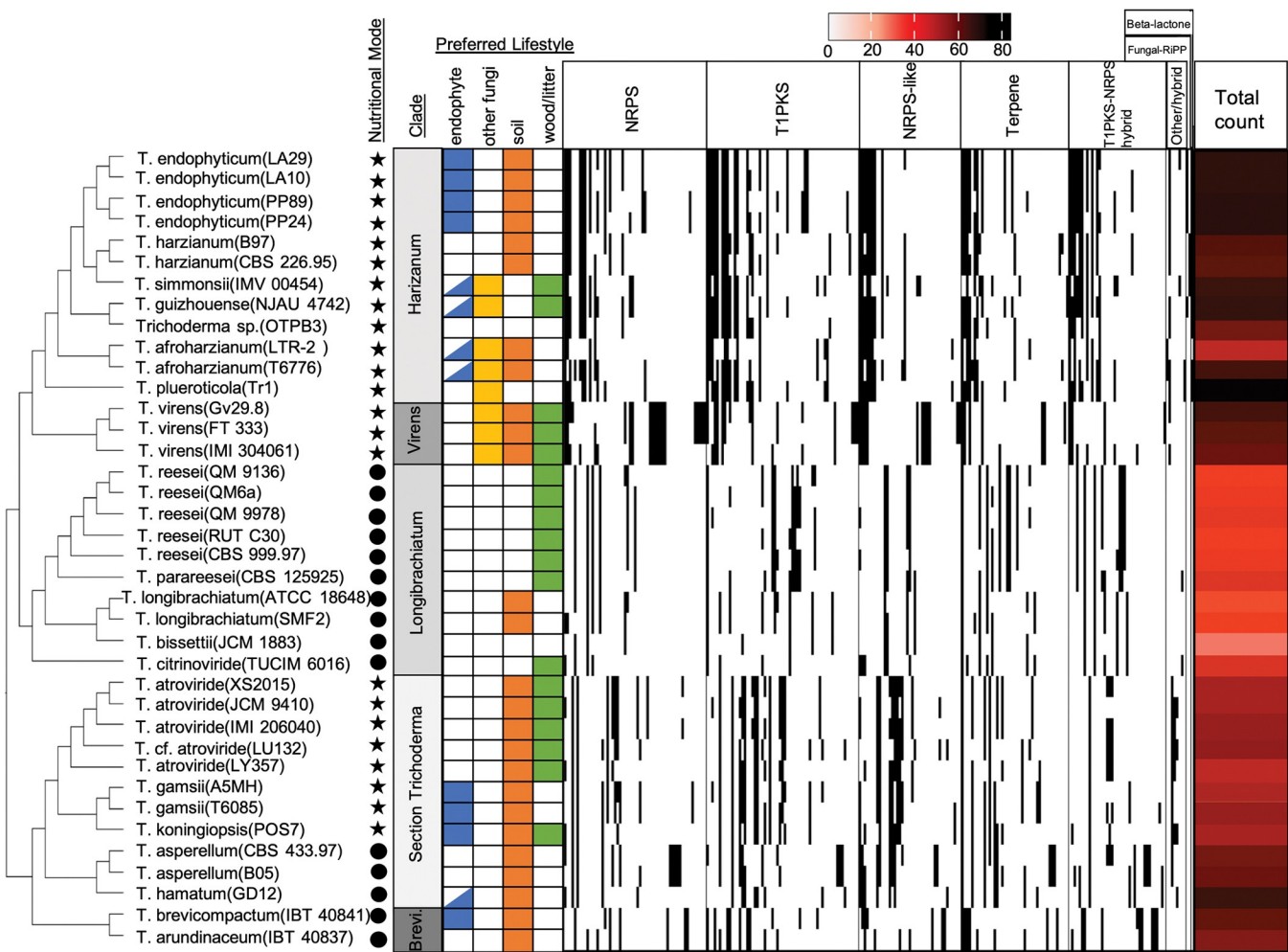

**Fig 2. Distribution and total amount of BGC families per genome varies across isolates and *Trichoderma* clade.** The presence of different BGC families are indicated by a black cell, BGC families are separated by class. The nutritional mode of each species is indicated by either a circle (saprotroph) or a star (mycotroph). BGC families present only in a single genome were removed for visualization purposes. *Trichoderma* species lifestyle is indicated by the presence or absence of a color-coordinated cell. Nutritional mode is indicated by either a circle (saprotroph) or a star (mycotroph). Endophyte cells half-filled indicated a potential endophytic lifestyle for that species. The total amount of DGC families identified in each genome are indicated by the red heatmap. Clade "Brevicompactum" is abbreviated as "Brevi".

## Several mycoparasitism genes are associated with the *Trichoderma* endophytic lifestyle

The *Trichoderma* genomes differ in their diversity and distribution of the 746 genes previously linked to fungal mycoparasitism [65–67]. Of the 746 mycoparasitism genes, 468 of these genes are present (at least one ortholog) in all 38 *Trichoderma* genomes, and some mycoparasitism genes have more than one ortholog present in a single genome. The number of total mycoparasitism genes per genome ranges from 676 (*T. longibrachiatum* ATCC18648) to 991 (*T. virens* IMI304061) and on average, a *Trichoderma* genome contains 850.4 mycoparasitism genes. The total amount of genes per genome positively correlated with the amount of identified mycoparasitism genes per genome (S15A Fig). Larger genomes tend to have greater numbers of mycoparasitism genes (S15B Fig) and genome quality (N50) does not correlate with the number of mycoparasitism genes identified per genome (S15C Fig).

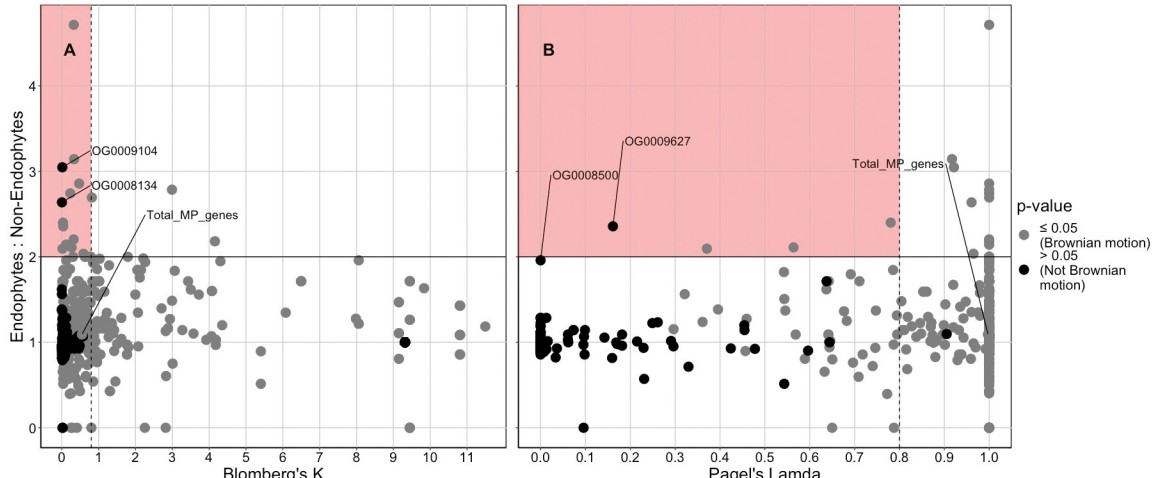

**Fig 3. Several mycoparasitism genes potentially related to the *Trichoderma* endophytic lifestyle were identified.** The phylogenetic and ecological (endophyte:non-endophyte ratio) signal of the distribution of the different mycoparasitism genes were determined by two methods. (A) Calculation of Blomberg's K for each mycoparasitism gene distribution suggests that only two mycoparasitism genes are both overdispersed in the *Trichoderma* phylogeny (K < 0.8) and is not consistent with Brownian motion evolution (p > 0.05), while also being overrepresented in endophyte genomes (E:NE > 2). (B) Calculation of Pagel's Lambda for each mycoparasitism gene distribution suggests that only a single mycoparasitism gene is both overdispersed in the *Trichoderma* phylogeny (Lambda < 0.8) and is not consistent with Brownian motion evolution (p > 0.05), while also being overrepresented in endophyte genomes (E:NE > 2).

The total diversity of mycoparasitism genes is significantly higher in endophyte genomes compared to non-endophyte genomes (S7D Fig), and mycotroph genomes have greater amounts than saprotroph genomes (S16 Fig). However, when phylogenetic signal was controlled for, the distribution and diversity of the total number of mycoparasitism genes as well the distribution of most individual mycoparasitism genes can be explained by *Trichoderma* phylogeny, as determined by Blomberg's K and Pagel's Lambda analyses (Fig 3A, 3B). There are three total orthogroups whose distribution is not consistent with Brownian motion evolution according to the two separate phylogenetic signal metrics and are also found to be overrepresented in endophyte genomes (Table 2). The two mycoparasitism genes orthogroups of interest from the Blomberg's K calculations are OG0009104 (function: unknown) and OG0008134 (function: unknown). The singular mycoparasitism gene of interest from the calculation of Pagel's Lambda is OG0009627 (function: unknown). From this same set of calculations, OG0008500 (function: unknown) was nearly considered an orthogroup of interest but was discarded due to its E:NE value (E:NE < 2). Besides the three genes discussed above, the remaining genes are not overrepresented in *Trichoderma* and have distributions consistent with Brownian motion (K > 0.8 or Lambda > 0.8, p ≤ 0.05). Most of the mycoparasitism genes enriched in endophyte genomes do not have a known eggnog functional annotation. The genes found in

**Table 2. Mycoparasitism gene orthogroups of interest that are found in greater numbers in endophytic *Trichoderma* genomes.**

| Gene orthogroup | Pagel's Lambda | Pagel's Lambda p-value | Blomberg's K | Blomberg's K p-value | M:S* | E:NE** | Eggnog Annotation |
|---|---|---|---|---|---|---|---|
| OG0009104 | 0.921 | 5.62E-08 | **0.014** | **0.077** | 3.048 | 4.783 | Function unknown |
| OG0008134 | 0.961 | 1.59E-07 | **0.004** | **0.239** | 2.637 | 6.522 | Function unknown |
| OG0009627 | **0.161** | **0.211** | 0.039 | 0.006 | 2.357 | 2.446 | Function unknown |

*Ratio of average mycoparasitism gene count in mycotrophic to saprotrophic *Trichoderma* genomes.

**Ratio of average mycoparasitism gene count in endophytic to non-endophytic *Trichoderma* genomes.

the highest ratio in mycotroph genomes vs. saprotroph genomes, as well as genes only found in mycotroph genomes, are shown in S4 Table (full results in S17 Fig). Similar to the endophyte results, most mycoparasitism genes enriched in mycotroph genomes or found only in mycotroph genomes had an eggnog functional annotation of "unknown". Mycoparasitism orthogroup OG16824 which does have an available functional annotation (function: Post-translational modification, protein turnover, and chaperones) was found only in mycotroph genomes and did not have a distribution consistent with Brownian motion (S4 Table).

Based on the total number of mycoparasitism genes per genome, the distribution of mycoparasitism genes varies greatly across the *Trichoderma* genus (S18 Fig). Longibrachiatum has the fewest of our targeted mycoparasitism-related genes, while Virens, section *Trichoderma*, and Harzianum have the most (S18A Fig). However, Brevicompactum and section *Trichoderma* have the highest percent of their genomes composed of mycoparasitism genes (S18B Fig). We ordinated the mycoparasitism gene count data after correcting for the phylogenetic signal inherently present the *Trichoderma* dataset using phylogenetic principal components analysis (pPCA), the results of which suggest that the number and distribution of the mycoparasitism genes primarily reflect *Trichoderma* phylogeny (S19 and S20 Figs). The differences in mycoparasitism gene distributions between the *Trichoderma* clades can clearly be seen in a visualization of the global positive and negative principal component values calculated from the pPCA (S19A Fig).

The *T. endophyticum* isolates have almost identical distribution and diversity of mycoparasitism genes compared to each other, with the total count ranging from 891 to 901. The average number of *T. endophyticum* mycoparasitism genes (897.0 genes) was comparable to the combined average in all other endophyte genomes (898.6 genes) and below the combined average in all other mycotroph genomes (910 genes). The phylogenetically adjusted PCA (pPCA) groups *T. endophyticum* genomes with their relatives in Harzianum (S20 Fig). No mycoparasitism genes were exclusively present or absent in the *T. endophyticum* genomes.

## Chitinase gene repertoires reflect vertical inheritance and nutritional mode, but not lifestyle in *Trichoderma*

Chitinases are a complex group of carbohydrate-active enzymes important for mycoparasitism, so we profiled chitinase repertoires across all genomes separate from the mycoparasitism genes analysis. The chitinase genes in our dataset are grouped into nine chitinase classes as delimited in [68], including classes AII (76), AIV (79), AV (107), BI (215), BII (105), BV (77), CI (107), CII (194), and ChitD (38), which collectively bear 153 lysin domains (LysM) and 374 chitin-binding domains (CBD) (Fig 4). Each *Trichoderma* genome contains two AII chitinases and one ChitD chitinase gene each. We found no class AIII, BIII, or BIV chitinase genes in any *Trichoderma* genomes. The distribution of chitinase genes and overall chitinase diversity varies greatly over our dataset and reflects the *Trichoderma* phylogeny (Fig 4). Chitinase diversity (total number of chitinase genes) within a genome ranges from 18 to 40 genes. Larger genomes tend to have a greater diversity of chitinase genes as well as a greater proportion of chitinase genes compared to total gene counts (S21A, S21B Fig). There was no correlation between genome quality (N50) and the total number of chitinase genes identified (S21C Fig).

Endophytic genomes did not contain a greater diversity of total chitinase genes compared to non-endophyte genomes (S7C Fig), or have a different total number of individual chitinases classes (S22A Fig). However, endophytic genomes did contain a higher percentage of class BII chitinase genes in their genomes as compared to non-endophytic genomes (S22B Fig). Blomberg's K and Pagel's Lambda analysis suggests that the distribution of all chitinase gene classes, including BII, are either consistent with the *Trichoderma* phylogeny, not overdispersed in the

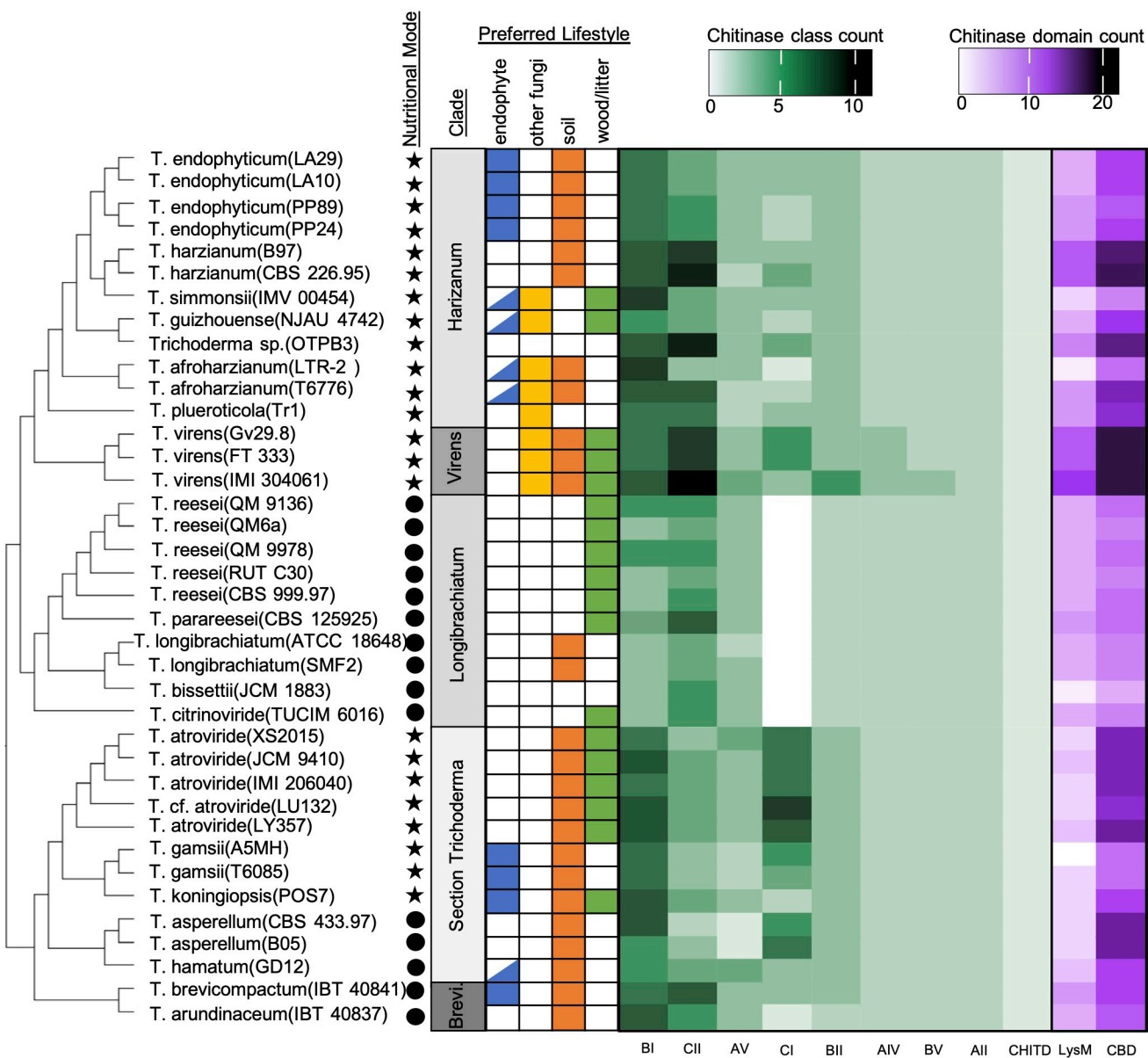

**Fig 4. Distribution and amount of chitinase gene classes per genome varies across isolates and *Trichoderma* clade.** Heatmap indicating the relative numbers of chitinase classes (green heatmaps) and chitinase domains (purple heatmaps) found in the genomes of 38 different *Trichoderma* isolates. Half-filled cells indicated a potential endophytic lifestyle for that species based on sequence identity and isolation in GenBank. Species lifestyle is indicated by color shading in cells, nutritional mode is indicated by either a circle (saprotroph) or a star (mycotroph). Clade "Brevicompactum" is abbreviated as "Brevi".

phylogeny, or not overrepresented in endophyte genomes (S23A, S23B Fig). Mycotrophic genomes contain a significantly greater amount of chitinase genes overall (S16C Fig) as compared to saprotrophic genome, and mycotroph genomes contain greater numbers of BI, BII, and CI chitinase genes (S24A Fig). Calculation of Blomberg's K and Pagel's lambda for the chitinase class counts and total chitinase gene count suggests that, similar to the endophyte: non-endophyte genome comparison, the number of total chitinase genes per genome is consistent with the *Trichoderma* phylogeny (S25 Fig).

There are significant differences in the diversity and proportions of chitinase genes to the total number of genes among the different *Trichoderma* clades (S26A, S26B Fig). Virens, Brevicompactum, and section *Trichoderma* contain the highest proportions of chitinase genes, and Longibrachiatum and Harzianum contain the lowest (S26B Fig). CI chitinase genes are completely absent in Longibrachiatum, while all other *Trichoderma* genomes contain at least one CI gene (Fig 4). Chitinase repertoires are distinguished by the presence/absence of genes that differ in the number of lysin domains (LysM) or chitin-binding domains (CBDs); the percent of LysM domains per chitinase gene is lowest in section *Trichoderma* (S27C Fig), and the percent of CBD per chitinase gene is highest in Virens (S27D Fig). The variation in LysM domains corresponds to the distribution of LysM-containing CII chitinases, which are fewer in section *Trichoderma* (Fig 4).

The *T. endophyticum* genomes have chitinase repertoires similar to other species in Harzianum (Fig 4). The four *T. endophyticum* genomes have an average of 26 chitinase genes each, with a near-identical distribution of these genes between the four genomes. Isolates PP24 and PP89 contain two CI and five CII chitinases, while LA10 and LA29 contain three CI and four CII chitinases, respectively (Fig 4). Domain architecture among the 26 chitinases in *T. endophyticum* strain PP24 is diverse, even among closely related genes (S28 Fig). The number of LysMs and CBDs are variable within chitinase classes, and a single member of the BV chitinase clade was inferred to contain an RTA1 Superfamily domain. LysM domains are present in all CII chitinase genes, with either one or two copies per gene. The *T. endophyticum* chitinases collectively contain fewer LysM domains than their closest relatives, the two *T. harzianum* genomes, primarily due to the higher number of LysM-containing CII chitinases in the *T. harzianum* genomes (Fig 4). CBD is present as a single copy in all *T. endophyticum* PP24 CII genes, and a single copy per gene is present in some but not all BI and BII genes (S28 Fig).

## Xylariales-like ergot alkaloid BGCs are found in *Trichoderma* species

In the course of identifying known BGCs across *Trichoderma* using antiSMASH's knownclusterblast [69] module, we incidentally noted a cluster of 11 homologs of ergot alkaloid BGC genes (cluster average bit score 1027.5, average amino acid identity 55.75%) in *T. brevicompactum* IBT40841 and *T. arundinaceum* IBT40837 (Brevicompactum clade, S5 Table). These clusters include a hypothetical protein and 11 ergot alkaloid biosynthesis homologs (UniProt): elymoclavine monooxygenase (*cloA*), chanoclavine synthase catalase (*easC*), lysergyl peptide synthetase subunits 1 and 2 (*lps1*, *lps2*), chanoclavine-I dehydrogenase (*easD*), chanoclavine-I aldehyde oxidoreductase (*easA*), argoclavine dehydrogenase (*easG*), chanoclavine-I synthase oxidoreductase *(easE)*, dimethylallyltryptophan N-methyltransferase (*easF*), dimethylallytryptophan synthase (*dmaW*), and a dioxygenase/oxygenase (*easH*). The hypothetical protein is not reported from any other ergot alkaloid BGC and is annotated as a nuclear transport factor 2 in InterPro or small polyketide cyclase (SnoaL) in UniProt [70].

To determine the evolutionary origin of the unexpected ergot alkaloid BGCs, we conducted a phylogenetic analysis of each gene in *T. brevicompactum's* BGC, including homologs recovered from Eurotiales (Eurotiomycetes, Ascomycota), Helotiales (Leotiomycetes, Ascomycota), Hypocreales (Sordariomycetes, Ascomycota), Onygenales (Eurotiomycetes, Ascomycota), and Xylariales (Sordariomycetes, Ascomycota) genomes (Fig 5). Eight of twelve ergot alkaloid BGC gene phylogenies highly support (> 95% IQ-TREE ultrafast BP) two Hypocreales clades: Hypocreales I, comprised of Clavicipitaceae species, and Hypocreales II, the Brevicompactum clade of *Trichoderma*, which is within a clade of predominantly Xylariales species. Only *cloA* supports (100% ultrafast BP) a single Hypocreales clade. *Aspergillus coremiiformis* CBS555.377 (Eurotiales) and *Pseudotulostoma volvatum* 6Q3QHG6JMA (Eurotiales) homologs are

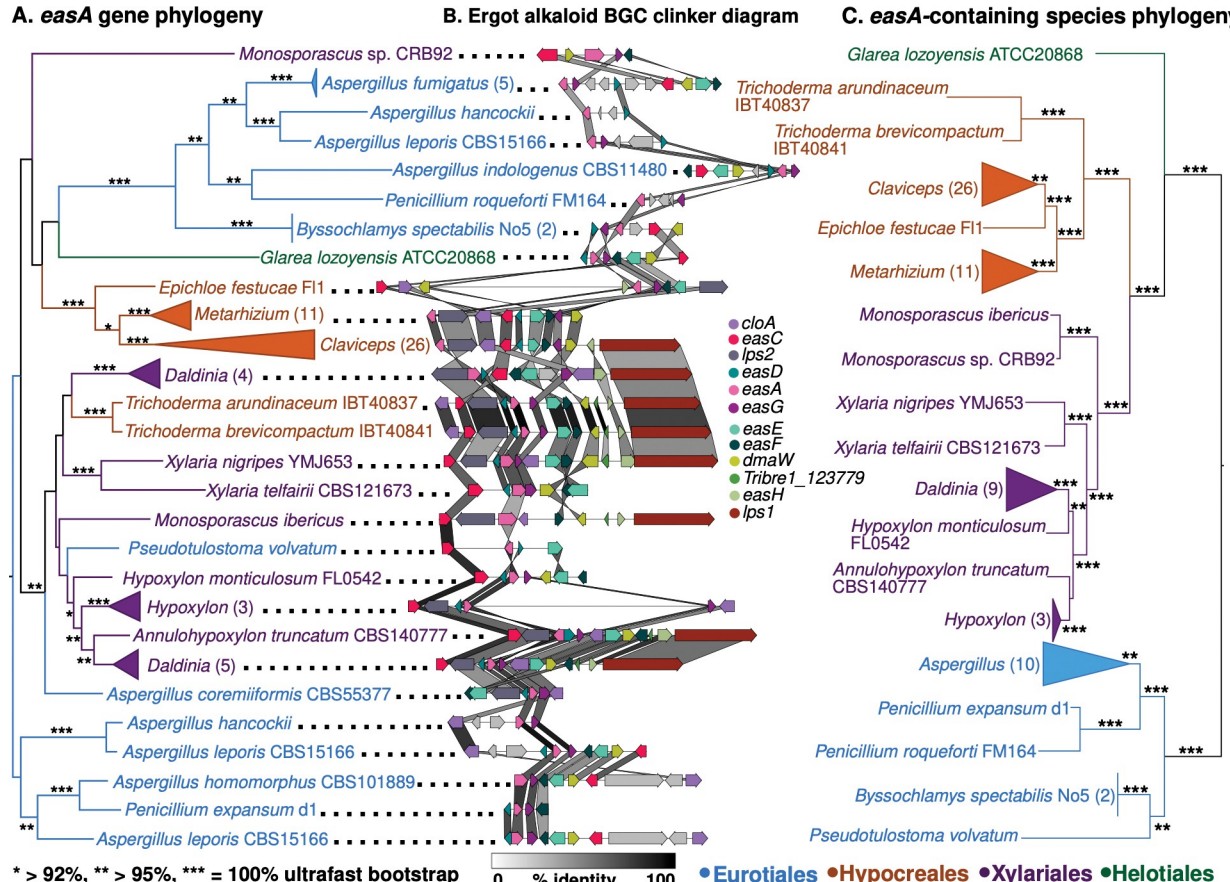

**A. *easA* gene phylogeny**

**B. Ergot alkaloid BGC clinker diagram**

**C. *easA*-containing species phylogeny**

\* > 92%, \*\* > 95%, \*\*\* = 100% ultrafast bootstrap

0  % identity  100

●Eurotiales  ●Hypocreales  ●Xylariales  ●Helotiales

**Fig 5. Horizontal transfer of ergot alkaloid BGCs from Xylariales to Brevicompactum and Eurotiales.** (A) IQ-TREE maximum likelihood gene phylogeny of *easA* with ultrafast bootstrap branch support. (B) Shared synteny and sequence similarity of ergot alkaloid BGCs. Gene-gene percent identity is highlighted in connecting shade. (C) Species phylogeny of genomes in *easA* gene phylogeny. Tip labels in (A) and (C) are colored by taxonomic order.

recovered with Brevicompactum clade and Xylariales species in 9 and 3 gene phylogenies respectively. All other homologs from Eurotiales species are placed on different branches in each gene phylogeny. We only recovered homologs of *easC* and a hypothetical protein (Tribre_123779) from other *Trichoderma* species. The *easC* homologs in other *Trichoderma* are part of a clade of non-ergot alkaloid BGC *easC* homologs and a *T. brevicompactum easC* paralog (100% ultrafast BP) that is distal to the cluster. The Tribre1_123779 phylogeny places the Brevicompactum clade and other *Trichoderma* homologs in two distinct subclades nested within Xylariales species homologs. We compared alternative hypotheses versus a HGT event by creating constrained *easA* phylogenies which forced monophyletic Xylariales, Hypocreales, or Eurotiales. These were all significantly worse than the optimal topology (S7 Table) according to an approximately unbiased test [71].

The gene order in the *Xylaria nigripes* YMJ653 (Xylariales) ergot alkaloid BGC is most similar to those in Brevicompactum genomes (mean protein amino acid identity 58.5%). Eleven of twelve syntenic homologs are in the same relative position in *Xylaria nigripes* YMJ653 and *T. brevicompactum*, whereas Clavicipitaceae (Hypocreales) species BGCs have a largely different gene order (mean protein amino acid identity 53.5%). Additionally, Tribre1_123779 is located in a clade of Xylariales homologs, but we did not recover homologs in Clavicipitaceae species.

The homologs of Tribre1_123779 in other *Trichoderma*, which lack ergot alkaloid BGCs, are part of a two-gene NRPS cluster they share with *Annulohypoxylon maetangense* CBS123835 (Xylariales).

## Discussion

The growing resource of publicly available *Trichoderma* genomes makes it possible to search for genes and genomic features associated with particular *Trichoderma* lifestyles, including the endophytic lifestyle [72]. Yet despite the prevalence of *Trichoderma* as endophytes [12, 23, 73, 74], few sequenced *Trichoderma* genomes were originally isolated as endophytes.

### *Trichoderma* species with a recorded endophytic lifestyle have a greater metabolic potential than non-endophytic species

Using the diversity of DGCs and BGCs in a genome as a proxy for the metabolic potential of *Trichoderma* species, we found that endophytic *Trichoderma* species have greater metabolic potential than their soil- or litter/wood-dwelling relatives. However, these results are difficult to disentangle due to the relationship between lifestyle and phylogeny within the *Trichoderma* genus, as well as the complexity inherent in the endophytic fungal lifestyle. The relative enrichment of metabolic potential in the endophytic *Trichoderma* species could result from unique selection pressures on endophytic fungi, a consequence of the natural complexity of the endophytic lifestyle or some combination thereof. Our findings, along with previous reports [26], contribute to the hypothesis that a general trend of increased metabolic capabilities is associated with the fungal endophytic lifestyle.

One potential explanation for the relative enrichment of DGCs in endophytes is the expanse of host plant chemical defenses that endophytic fungi must resist to persist inside the host plant. There is conflicting evidence about the response of the host plant to the colonization by an endophyte. Some evidence suggests that certain endophytic fungi elicit different defense responses compared to those triggered by pathogenic fungi [75, 76]. Other evidence suggests that upon the initial colonization, plant receptors indiscriminately identify both pathogenic fungi and endophytic fungi as intruders [77–80], triggering plant immune responses which include the production of defensive secondary metabolites. These chemical defenses include aromatic compounds and phenolics [81], which can negatively impact fungal growth and survival [82, 83]. Despite being a potential benefit to the host plant, the endophyte must contend with the same profile of defensive metabolites that restrict potential pathogens [84]. Although plant-produced secondary metabolites are present in soil, litter, and wood and therefore influence microbial growth and activity [85, 86], higher concentrations of induced defensive metabolites in the plant may provide focused selection for their degradation by endophytic fungi, as previous research suggest that metabolite-degrading gene clusters distribution is shaped by both fungal phylogeny and ecology [87]. Additionally, endophyte species that are horizontally transmitted and are potentially exposed to defensive metabolites from diverse hosts may benefit from a greater diversity of DGCs. A more diverse repertoire of DGCs may promote endophyte survival in the host plant by improving the breadth of host plant defensive compounds the endophyte can degrade.

The unique pressures presented by the host plant may explain in part the pattern of DGC presence and absence in the *Trichoderma* dataset. The elevated DGC diversity in endophytic *Trichoderma* species is underpinned by increased numbers of SAH, ARD, PMO, and FAD DGCs, which is consistent with previous research suggesting that fungal endophyte genomes are enriched in ferulic acid decarboxylase (FAD) DGCs, and plant symbiotic fungal genomes are enriched in DGCs containing aromatic ring-opening dioxygenase (ARD) (Gluck-Thaler

and Slot, 2018). Living plants contain a variety of aromatics and flavonoids, which are either constitutively expressed or induced as plant defenses against pathogens, pests, and herbivores [81, 88, 89]. Given that these metabolites are the likely targets of ARD and NAD DGCs, a higher diversity of these DGCs may contribute to fungal endophytes' success in persisting within the host plant. However, endophytic species not in Harzianum did not contain ARD (Fig 1), suggesting that ARD may not be essential for an endophytic lifestyle. The Longibrachiatum clade, which contains only species found on plant litter/soil and/or wood, was the only *Trichoderma* clade to completely lack NAD DGCs (Fig 1), consistent with these fungi not often being exposed to these compounds in their habitat. No DGCs containing vanillyl alcohol oxidases were detected in any of the *Trichoderma* genomes (Fig 1). This class of DGC was previously found to be enriched in soil-dwelling, saprotrophic genomes, which may benefit from lignin decay enzymes (Gluck-Thaler and Slot 2018), suggesting that *Trichoderma* species may avoid or utilize other mechanisms to tolerate lignin byproducts.

The higher diversity of BGCs found in endophytes may allow them to overcome the unique pressures of an endophytic lifestyle. For example, metabolites that mimic plant hormones may facilitate communion with the host plant or produce plant growth-promoting compounds [17]. Some plant growth-regulating compounds produced by *Trichoderma* species are harzianolide [90–92], harzianic acid [93], and 6-pentyl-2H-pyran-2-one [94, 95]. BGCs in *Trichoderma* species may also produce siderophores that sequester iron [50, 96] or metabolites that inhibit antagonistic microbes [97–99], or in the case of harzianic acid, both iron sequestration and antifungal ability [100]. Although the majority of *Trichoderma* secondary metabolite products have not been linked to their respective BGCs [72], a greater diversity of BGCs is expected to facilitate a wider variety of interactions between a fungus and its surrounding environment, potential hosts, and other microorganisms.

Due to "cross-talk" between plant and endophyte metabolic pathways, a diverse metabolic gene cluster repertoire may lead to even further diversification of the metabolite profile in the holobiont [101]. *Cola acuminata* produces camptothecin, and a common endophyte of this plant (*Fusarium solani)* produces camptothecin precursors [102, 103]. In this system, the fungus is missing a key enzyme that is present in the plant that allows the production of camptothecin. However, since the plant can transform the fungal metabolites, this increases the availability of camptothecin in the plant. The opposite of this case is also true, as endophytic fungi are known to modify terpene metabolites produced by their host plant [104]. These observations are consistent with the "mosaic effect" theory, which states that endophytes create a variable metabolite composition within and between plant organs that would otherwise contain a uniform profile of metabolites, thereby increasing the defense ability of the plant [105].

Increased diversity of both DGCs and BGCs in endophytes may ultimately improve the fitness of the holobiont (a plant and all closely-associated microorganisms) by contributing to the overall collection of defensive metabolites available to the plant and acting as the host plant's "acquired immune system" [97]. Endophytes increase host availability in the ecosystem by increasing host protection from pathogens and herbivores. Endophytes can produce novel metabolites in the host plant or produce similar metabolites already synthesized by the host plant [102, 106, 107]. Fungal endophytes of several grass species (e.g., *Epichloë* spp.) produce a variety of novel defense alkaloids which can directly protect the host plant by reducing feeding by both invertebrate and vertebrate herbivores [108, 109]. *Trichoderma* species specifically are known to produce a multitude of fungal defensive secondary metabolites beneficial to the host plant [98, 110], such as the antifungals trichodermin and harzianopyridone [90, 111]. Fungal endophytes may also influence the blend of volatile organic compounds exuded by the plant, which attract predators of the herbivorous aphid species, thereby decreasing herbivory of the host plant [112]. Taken together, our findings suggest that the selective pressure for increased

overall fitness of the holobiont via metabolite diversity may select for more metabolically diverse pathways in endophytic fungal genomes.

Alternatively, elevated metabolic diversity in endophytic genomes may be associated with these fungi being able to exploit multiple lifestyles throughout their lifecycle [26, 113–115], including utilizing decaying plant biomass [116–118]. Indeed, each of the endophytic *Trichoderma* isolates in our dataset are more or less "ecological generalists" that have been associated with multiple lifestyles (S1 Table). Fungi with more complex interactions with other organisms, such as plants or other fungi, tend to produce a greater diversity of bioactive compounds [119]. For example, *T. virens*, capable of multiple lifestyles (wood/litter-, soil-, and fungi-dwelling), is well-known for producing a variety of bioactive metabolites [50, 65, 120]. In this study we also observed that genomes of *Trichoderma* species with multiple potential lifestyles have greater numbers of DGCs and BGCs than genomes of *Trichoderma* species only exhibiting a single lifestyle (S29A, S29B Fig). In contrast, metabolic gene cluster diversity is reduced in Longibrachiatum, which contains exclusively single-lifestyle species (S11A, S11B Fig).

Rather than an increase in metabolic diversity in multi-lifestyle genomes, there may have been a past reduction in diversity in single-lifestyle genomes. Species in Longibrachiatum were shown to be able to degrade comparatively fewer metabolites in laboratory metabolite digestion challenges compared to species in Harzianum or other plant-related *Trichoderma* species [5], consistent with our findings of their metabolic potential (S11A and S29A Figs). Specifically, all species in Longibrachiatum completely lack ARD and NAD DGCs (Fig 1), despite these species living exclusively in wood or plant litter where aromatic compounds are present as products of wood decay. Previous studies on the succession of fungal species in leaf litter and dead wood indicate that *Trichoderma* species are more abundantly found as late-term colonizers, active only after other microbe community members have degraded or modified complex compounds such as lignin or complex metabolites [121–125]. By depending on other members of the substrate microbiome to degrade aromatic compounds [81, 126], litter- and soil-dwelling *Trichoderma* species would not be under the same selective pressure as endophytic species to retain DGCs, resulting in the loss of these DGCs. Metabolic diversity may reinforce ecological generalism in some lineages, while specialized lifecycles in others favor metabolic streamlining.

The trends observed in the metabolic capabilities of endophytic *Trichoderma* are exemplified by newly sequenced *T. endophyticum* genomes isolated from tropical trees. This study shows that *T. endophyticum* is an extreme example of the characteristics typically exhibited by endophytic *Trichoderma* species. The four *T. endophyticum* genomes are among the genomes with the greatest numbers of DGCs, BGCs, and metabolic genes as compared to their close relatives (S2 Fig). Previous comparative analyses suggest that endophyte genomes tend to have comparatively larger genomes [26], a trait that was also observed in our four *T. endophyticum* genomes (S4 Fig). The four *T. endophyticum* genomes were above the trend comparing number of DGCs and BGCs to genome length (S4A, S4B Fig) and the proportion of DGCs and BGCs to the total gene number (S5 Fig) and exceeded their closest relatives in the Harzianum clade in DGC measures (S4A and S5A Figs). *Trichoderma endophyticum* may be of interest in natural product discovery due to the comparative richness of its metabolic diversity [127, 128]. However, this study only evaluated four *T. endophyticum* genomes from two geographic locations and two host plant species. Given that *T. endophyticum* isolates are more similar based on geographic location and host plant species [3], more representatives of this species should be sequenced from different host tree species and from a wider geographical range in order to understand more about the intra-species diversity of *T. endophyticum*.

## Mycoparasitism-related genes

Each of the 38 *Trichoderma* genomes contains multiple genes in orthogroups made from our list of 746 key mycoparasitism genes. The distributions of the mycoparasitism gene repertoires are consistent with phylogenetic relationships. There was no indication of grouping based on fungal lifestyle, even after adjusting for phylogenetic relationships within the dataset (S20 Fig). Although the main indicator of mycoparasitism gene diversity and distribution was phylogeny, mycotrophic genomes contained significantly greater numbers of the mycoparasitism-related genes in mycotrophic genomes as compared to saprotroph genomes (S16D Fig).

Overall, endophytic *Trichoderma* genomes had greater abundances of mycoparasitism-related genes than non-endophytic genomes (S7D Fig). However, this difference was not as distinct as the difference in abundances between mycotrophic and saprotrophic genomes (S16D Fig). Potentially, the relative abundance of mycoparasitism genes could be due to some of the mycoparasitism genes being beneficial for creating or maintaining a host-endophyte symbiosis. For instance, a gene encoding an aspartyl protease (OG0009580) was overrepresented in both mycotrophic and endophytic genomes (S3 Table); aspartyl proteases are cell wall degrading enzymes known to be produced upon competition of a fungal target [129] as well as upon colonization of plant host [130]. The ability to digest a plant cell wall would be positively selected for horizontally-transmitted endophyte species, as this ability would increase the chances of reaching the inside of the plant. Our results support similar findings, which suggest that certain cell-wall degrading enzymes are overrepresented or expanded in fungal endophytes, also theorized to be useful for the initial infection and colonization of the host plant [131]. Alternatively, the relative abundance of mycoparasitism genes in endophytic genomes may simply be due to the evolutionary history of *Trichoderma*, which is hypothesized to have a mycoparasitic origin [4], as the mycoparasitic nutritional mode is widespread throughout this genus and most species with a recorded endophytic lifestyle also have a mycotrophic nutritional mode.

## Chitinase prevalence and other key mycoparasitism-related genes are not correlated with an endophytic lifestyle in *Trichoderma*

Each *Trichoderma* genome contained one or more chitinase genes of different classes (Fig 4). Mycotrophic genomes contained a significantly greater amount of total chitinase genes as compared to saprotrophic genomes (S16C Fig). Mycotrophic genomes contained greater numbers of BI, BII, and CI chitinases (S24A Fig). The relative abundance of chitinases in mycotrophic genomes is not unexpected since chitinase production is key to mycoparasitism, many mycoparasitic *Trichoderma* species are well-known chitinases producers [132], and as previously stated, mycoparasitism is the ancestral state of *Trichoderma* [50]. Similar to the results for metabolite gene clusters and mycoparasitism gene repertoires, we also determined that the overall pattern of chitinase gene distribution was consistent with *Trichoderma* phylogeny (S25A, S25B Fig).

One of the benefits endophyte confer to their host plant is direct competition of invading fungi via mycoparasitism [133, 134]. Due to this selective pressure for mycoparasitic ability in endophyte genomes, we investigated if endophytic *Trichoderma* species have expanded chitinase repertoires. However, we did not find any difference in the overall number of chitinase genes (S7D Fig) or individual number of chitinase gene classes (S22A Fig) between endophytic genomes and non-endophytic genomes. Only when we looked at percent of total gene number did we identify any differences in endophytic and non-endophytic chitinases content; the percent of chitinase BII genes in each genome is significantly greater in endophytes (S22B Fig). This relative increase in BII chitinases could simply be due to the mycoparasitic history of

*Trichoderma* [50], because we did not identify a significant signal associated with an endophytic or mycoparasitic lifestyle when we controlled for phylogenetic signal (S23A, S23B Fig).

## Horizontal transfer of an ergot alkaloid cluster is consistent with a defensive role in endophytes

BGC repertoire diversification in *Trichoderma* is in part shaped by horizontal gene transfer between Eurotiales, Hypocreales, and Xylariales. Given the precedent of horizontal transfer in Hypocreales and Brevicompactum clade [135–137], we were not entirely surprised to identify BGC acquisitions among *Trichoderma* species, including the ergot alkaloid BGC [138]. The species phylogeny places Clavicipitaceae and *Trichoderma* within Hypocreales. However, eight of twelve ergot alkaloid gene phylogenies support Brevicompactum's placement within a Xylariales-dominant clade (> 95% ultrafast BP per gene) apart from the Clavicipitaceae clusters. Topology testing rejected constraints that forced the *easA* phylogeny to be consistent with vertical inheritance (S7 Table). HGT is supported by the shared gene order between *Xylaria nigripes* YMJ653 and *T. brevicompactum*, which differs from the gene order in Clavicipitaceae. Additionally, we recovered a novel ergot alkaloid gene family (Tribre1_123779) that is shared between *T. brevicompactum* and Xylariales ergot alkaloid clusters but absent in Clavicipitaceae, further supporting the HGT. The ergot alkaloid BGC was likely obtained after Brevicompactum diverged from other *Trichoderma* because the only homologs we recovered in other *Trichoderma* are distant *easC* paralogs and independently transferred xenologs of Tribre1_123779. Interestingly, the independent transfers of Tribre1_123779 to other *Trichoderma* species are part of a different NRPS cluster acquired from Xylariales species that also lack the ergot alkaloid BGC (100% ultrafast BP). We further identified ergot alkaloid BGCs in two Eurotiales species that are more closely related to the Xylariales clusters and Brevicompactum clusters than other Eurotiales', suggesting there have also been inter-class HGTs involving Xylariales. These findings add to previous work that suggested the origin of the Clavicipitaceae (Hypocreales) ergot alkaloid BGC was by HGT from Eurotiales species [139]. HGT of metabolic gene clusters to *Trichoderma* has been previously reported [140], including a recent report of HGT of a PKS BGC to Brevicompactum from Eurotiales [138]. Our finding of multiple novel HGTs between Eurotiales, Xylariales, and *Trichoderma* (Hypocreales) provides evidence of a potential horizontal transfer highway between these orders.

HGT is thought to indicate shared selection pressures in the donor and recipient lineages and can suggest possible ecological roles of the products of the transferred BGC [34]. Ergot alkaloids are generally considered to have roles in animal-associated fungal niches [141]. They can act as anti-herbivory compounds in the plant endophytic/pathogenic *Claviceps* and *Epichloë* species [142]. They are also upregulated during insect colonization by *Metarhizium* insect pathogens [143]. The most closely related clusters to Brevicompactum's ergot alkaloid BGCs are also in animal-associated, plant-pathogenic, or endophytic Xylariales: *Daldinia childiae* (Xylariales), *D. decipiens* CBS113046, *D. loculata* AZ0526, *D. loculata* CBS113971, *Xylaria nigripes*, and *X. telfairii* CBS121673 [144–153]. All of these species have been isolated as endophytes or from decaying wood, while *D. decipiens* is also a symbiont of *Xiphydria* woodwasps [146], and *X. nigripes* is one of several termite-associated *Xylaria* species [154]. Xylanigripones are the first described ergot alkaloids from Xylariales and are produced by *X. nigripes* [155], and these compounds are the likely product of the *X. nigripes* cluster we identified. *X. nigripes* has an extensive history of medicinal consumption in humans attributed to its diverse pharmacological activity, including effects on the nervous system, which is typical of ergot alkaloids. Given the high similarity between Brevicompactum's ergot alkaloid BGCs and *X. nigripes*', it is likely that Brevicompactum species produce similarly structured compounds.

More broadly, the presence of novel ergot alkaloid BGCs throughout Xylariales suggests there may be undescribed ergot alkaloids from Brevicompactum and Xylariales.

The ecological function of xylanigripones is not known, nor are any shared ecological pressures that could drive HGT from Xylariales to Brevicompactum. This HGT may simply represent the potential for mycotrophs like *Trichoderma* species to acquire genetic material from hosts [37, 140]. Additionally, endophytic ecologies in Brevicompactum and Xylariales may facilitate HGT through shared habitat [156]. Indeed, the majority of Xylariales species with ergot alkaloid BGCs have been isolated as endophytes. Furthermore, the two Eurotiales clusters within the Xylariales ergot BGC donor clade are potential plant associates, including *Pseudotulostoma volvatum* [157], which is associated with tree roots [158] and *Aspergillus coremiiformis* CBS55377, an anomalous *Aspergillus* with a reduced genome, but maintenance of plant cell wall degradation enzymes [159]. We speculate that the putative products of ergot alkaloid BGCs might contribute to plant host defense in Brevicompactum endophytes, given their documented role in anti-herbivory and the widespread distribution of ergot alkaloid BGCs in Xylariales endophytes and animal-associated species.

### Limitations in analysis

It is important to note that there may be some error in assigning the gene clusters to particular cluster families due to fragmentation of the clusters or cluster core genes not being present in annotations. Fragmented secondary metabolite clusters are likely still recovered by antiSMASH if the core genes are present, and BiG-SCAPE's model attempts to account for cluster fragmentation in cluster family grouping. However, core genes that were not called in the annotation are not detected by antiSMASH, which is an ongoing problem in the field. For DGCs, we adjusted the weights of BiG-SCAPE's cluster relationship network, so these weights may not optimally group DGCs into families. To better account for fragmented and DGC grouping, we opted to use BiG-SCAPE clan designations, which are more crude.

Other limitations of this study include difficulties assigning lifestyles to the different *Trichoderma* species due to its ubiquitous nature and oftentimes cryptic nutritional modes and/or lifestyles [3, 4]. For these analyses, *Trichoderma* lifestyles were determined by finding two or more reports of confirmed *Trichoderma* species being isolated from the internal tissues of a living plant, as described in the "Assignment of ecological data to isolates" section in methods.

## Materials and methods

### *T. endophyticum* DNA extraction and sequencing

Four isolates of *T. endophyticum* were isolated from the sapwood of living rubber trees (*Hevea* spp.) from Peru. Permits for the collection of endophytic fungi in Peru were given by the Peruvian Department of Agriculture, permit number: 0093-2009-AG-DGFFS-DGEFFS. Samples were imported into U.S.A. under the Animal & Plant Health Inspection Service (APHIS) permit: P526P-08-03319-Label#7. Isolates were identified by Sanger sequencing the ITS region. Isolates were individually grown in Potato Dextrose Broth (PDB) in sterile flasks on a shaker plate for several days in darkness. Mycelium was filtered from the PDB and rinsed twice with 50m of sterile MiliQ water. The mycelium was frozen using liquid nitrogen and ground into fine dust using a mortar and pestle. We extracted DNA from the pulverized mycelium using the DNeasy Plant Minikit (Qiagen), which was then sequenced with both short (Illumina) and long-read (Oxford Nanopore Technology) technologies. We prepared the DNA for long-read sequencing using the Oxford Nanopore Technologies library preparation kit SQK-LSK108, using the standard kit instruction and G-Tube fragmentation. The prepared DNA was individually sequenced using the Nanopore MinION Mk1B and flow cell with R9.4.1 chemistry,

resulting in approximately 10x coverage of each *T. endophyticum* genome. Nanopore raw sequence data was basecalled with Albacore v.2.3.1. The Illumina PE150 sequencing was performed using a NovaSeq 6000 sequencer.

## Genome assembly

We retrieved publicly available raw SRA data from NCBI and JGI for 35 *Trichoderma* strains. For both the publicly available read data and our own *T. endophyticum* reads, we removed adapters and low-quality reads from the short-read data with Trimmomatic v0.36 [160] and assessed overall read quality with FastQC v0.11.7 [161]. We used SPAdes v3.12.0 [162] to assemble each *Trichoderma* genome, using both the Illumina and Nanopore reads to produce hybrid assemblies for the four *T. endophyticum* isolates. All assemblies had their quality metrics (e.g., N50) assessed with QUAST v4.6.3 [163] and coverage assessed with BBmap v37.93 [164].

## Genome annotation

All genomes were annotated using a MAKER v2.31.8 [165] annotation pipeline. Genome annotation for the entire 39-genome dataset was performed identically for each separate genome. For each genome, de novo repeat libraries were created with RepeatModeler [166]. Our MAKER2 pipeline was run with three iterations in order to fine-tune the genome annotations. In the first iteration, protein and EST data from the same or closely related *Trichoderma* species were retrieved from NCBI for each genome. The first iteration of MAKER used the repeat library and provided EST and protein data to generate individual preliminary gene annotations for each genome. For the second iteration of the MAKER2 pipeline, we provided *ab initio* gene predictions from SNAP v.2013-02-16 [167] (produced from preliminary gene predictions from the first iteration) and Augustus v.3.3 [168] (trained using BUSCO v.3.0.1 [169]), and GeneMark v4.32 [170]. SNAP and Augustus were both retrained using the high-quality predictions from iteration 2, and this refined dataset was supplied to MAKER2 for the final iteration to produce the final genome annotations for each isolate (option keep_preds = 1).

## Genome dataset curation

The complete dataset of 39 *Trichoderma* genomes includes ecologically and geographically diverse isolates. The genes encoding translation-elongation factor 1 alpha (TEF1) and RNA polymerase II gene (RPB2) sequence were identified in each genome and compared to their respective GenBank type specimen sequences using BLASTn. In the case of several *Trichoderma* isolates, the original species identification was determined to be incorrect, and the species was re-labeled. Available in S1 Table are details of each isolate's revised species identification, assembly source (public or re-assembled), nutritional mode, species lifestyle, and country of origin.

## Assignment of ecological data to isolates

Lifestyle and nutritional mode data for each species was collected using the same approach used in Chaverri and Samuels (2013). Briefly, lifestyle information (i.e., endophyte, on decaying plant material, in soil, and on other fungi) was based on published research, and nutritional mode (i.e., mycotrophy vs. saprotrophy) was based on ancestral character reconstructions, phylogenetic affinities, and confirmed antagonism experiments. Two *Trichoderma* strains (*T. bissettii* [JCM 1883] and unknown *Trichoderma* species [OTPB3]) were left

as "unknown" because no ecological information was available. For each *Trichoderma* species, we assigned a potential endophytic lifestyle by identifying isolates reportedly isolated from living plant tissue, documented on NCBI. To be a potential endophyte, we required at least two isolates from separate sources with a blastn match for one of the following: TEF1, 100% coverage and >99.5% identity, and RPB2, 100% coverage and >99.0% identity. For *T. arundinaceum*, there was no RPB2 sequence from type material available on NCBI, so we looked for matches with TEF1, 100% coverage and >99.5% identity and ITS 100% coverage and >99.5% identity.

## Ortholog supplementation and identification

To find orthologous sequences across the 39 genome set, we provided the final protein datasets for each *Trichoderma* genome to OrthoFinder v2.2.6 [171]. The resulting orthology datafiles and the MAKER protein files were processed with Orthofiller [172] to improve the genome annotations and identification of orthologs. All subsequent analyses used the post-Orthofiller protein datasets.

## Phylogenomic tree creation

Using the protein sequences of the single-copy orthogroups produced from OrthoFinder, we placed each orthogroup into an automated pipeline that built alignments for each orthogroup using mafft v7.407 [173], and alignments were automatically curated using the automated1 algorithm in TrimAI v1.4 [174]. RAxML was used for phylogenetic analyses [mapping percentage of 100 rapid bootstraps to the best-scoring ML tree], which resulted in 154 SCO trees with a median BP ≥98%. These 154 SCO trees were used to create the majority rule extended consensus tree using RAxML v8.2.11 (-m GTRCAT) [175]. We produced an alignment consensus phylogenomic tree from the 154 SCO IQ-TREEs with >98% BP support in IQ-TREE v2.0.3 (-bb 1000 -bsam GENESITE -m TEST) [176].

## Degradative gene cluster identification

We used a custom pipeline (https://github.com/egluckthaler/cluster_retrieve) [30] identify putative gene clusters associated with secondary metabolite degradation in the 39 annotated *Trichoderma* genomes. This custom script searched each genome for multiple gene cluster models containing particular "core" genes, or genes known to be important for processing secondary metabolites. We searched for gene clusters containing the following 13 core genes: aromatic ring-opening dioxygenase (ARD), benzoate 4-monooxygenase (BPH), ferulic acid esterase 7 (CAE), catechol dioxygenase (CCH), epicatechin laccase (ECL), ferulic acid decarboxylase (FAD), pterocarpan hydroxylase (PAH), naringenin 3-dioxygenase (NAD), phenol 2-monooxygenase (PMO), quinate 5-dehydrogenase (QDH), salicylate hydroxylase (SAH), stilbene dioxygenase (SDO), and vanillyl alcohol oxidase (VAO) (Gluck-Thaler and Slot 2018). Homologous genes at each locus were defined by a minimum BLASTp (v2.2.25+) bitscore of 50, 30% amino acid identity, and a target sequence alignment 50–150% of the query sequence length. Homologs of the query genes were considered clustered if a maximum of 6 intervening genes separated them; gene clusters on the same contig were consolidated if they were separated by less than 30kb. DGCs containing the same core gene are referred to as belonging to the same cluster class. When comparing DGCs across different genomes, DGCs exhibiting similar organization with a high degree of sequence similarity will be referred to as the same DGC family.

## Biosynthetic gene cluster identification

We predicted secondary metabolism clusters by submitting our *Trichoderma* annotations through the antiSMASH v5.0b [177] pipeline with glimmerhmm gene prediction enabled. Clusters were analyzed via the 'knownclusterblast' module to identify potential homologs of MIBiG v2.0 [69] database entries. As additional quality filters, we only referenced MIBiG accessions with 3 or more genes and removed knownclusterblast genes with bit scores lower than 100, redundant hits within each cluster, clusters with aggregate bit scores less than 100 x cluster gene quantity, and any clusters with less than 60% of the respective MIBiG cluster's gene quantity. We parsed and compiled data from the antiSMASH and knownclusterblast output using *smashStats.py* (gitlab.com/xonq/mycotools). BGCs are grouped in different classes respective to their association with different secondary metabolite compound classes, according to antiSMASH nomenclature (e.g. polyketides, non-ribosomal peptides, terpenes, indole etc.).

## Cluster family classification

We grouped gene clusters into homologous cluster families using BiG-SCAPE v1.0.1 [178]. Secondary metabolism clusters were classified into cluster families using default parameters while referencing MIBiG v2.0. For degradative clusters, we modified BiG-SCAPE's default model weights to 63% protein domain sequence similarity, 35% protein domain presence-absence, 2% conserved synteny, and a core gene boost set to 2x. *Trichoderma* cf. *atroviride* (LU140) produced a very fragmented genome which skewed the identification of MGCs, and as such, this genome was removed from the subsequent BGC and DGC cluster family designations. The differences in the overall counts of DGCs, BGCs, and their respective gene cluster classes between *Trichoderma* clades were determined using ANOVA or Kruskal-Wallis, and differences between both endophyte vs non-endophyte and mycotrophic vs saprotrophic were determined using a 2-sample t-test or Wilcoxon rank sum test.

## Mycoparasitism-related genes analysis

Using previously published results of differential gene expression analyses from RNA-seq and oligonucleotide tiling array data, we compiled a list of 746 genes from *T. atroviride*, *T. virens*, *T. reesei*, and *T. harzianum* that were upregulated during at least one time point or treatment during mycoparasitism assays compared with control time points or treatments [65–67]. For each of these 746 mycoparasitism-related genes, we identified the highest scoring hit in each *Trichoderma* genome using BLASTp (E value threshold = 0.001). Any orthogroup containing a highest scoring hit to a mycoparasitism-related gene was considered to be a mycoparasitism-related orthogroup. The differences in the overall counts of mycoparasitism-related genes between *Trichoderma* clades were determined using ANOVA or Kruskal-Wallis, and differences between both endophyte vs non-endophyte and mycotrophic vs saprotrophic were determined using a 2-sample t-test or Wilcoxon rank sum test.

## Chitinase analysis

We used an updated kingdom-level GH-18 chitinase ontology [68] to classify chitinases across the dataset. Chitinase sequences were first identified by hmmsearch for proteins containing Glyco_hydro_18 (PF00704) domain in hmmer-3.1b2 [179]. Chitinase genes from all *Trichoderma* species were then incorporated into a multifasta file with diverse representatives of each chitinase class in Goughenour et al. (2020) [68]. The sequences were then aligned in mafft v7.487 [173] and analyzed in FastTree v2.1.10 with default settings [180]. Sequence chitinase

classes were then manually assigned based on placement in the FastTree phylogeny. Domain architecture for *T. endophyticum* PP24 chitinases (S12 Fig) was determined using a CDD search [181] and mapped to an IQ-TREE (v. 1.6.11) phylogeny inferred under the LG+F+R3 model, which was determined by Model Test to be the best fit according to the BIC, with support computed by 1000 ultrafast BS replicates. The differences in the overall counts of chitinase gene classes between *Trichoderma* clades were determined using ANOVA or Kruskal-Wallis, and differences between both endophyte vs non-endophyte and mycotrophic vs saprotrophic were determined using a 2-sample t-test or Wilcoxon rank sum test.

## Phylogenetic correction: Fritz and Purvis' D, Blomberg's K, and Pagel's Lambda

To determine how the distribution and diversity of binary traits, such as the presence/absence of individual DGC and BGC families, are consistent with the phylogeny of *Trichoderma*, we calculated Fritz and Purvis' D statistic [182] for each individual gene cluster family using the "phylo.d" function from the "caper" v1.0.1 package [183] in R. Fritz and Purvis' D indicates the level to which species' shared phylogenetic history impact the presence or absence of any given trait. Fritz and Purvis' D Statistic is a measure of "phylogenetic signal" in the distribution of a trait. A D of zero indicates that a binary trait follows the Brownian motion model of evolution, where differences in distribution can be entirely explained by the phylogenetic history. When D = 1, this indicates that a binary trait is randomly distributed. D values above 1 indicate that traits are more overdispersed than expected by the Brownian motion model, and together with ecological distribution bias (see below) could suggest a trait is associated with particular lifestyle or nutritional mode. As D falls below 0, a trait is considered underdispersed (more conserved than the Brownian motion model). For the Fritz and Purvis's D-statistic, a significant p-value indicates that a particular gene cluster's distribution is significantly different compared to a simulated random distribution (a phylogeny under Brownian motion), therefore its distribution is not entirely explained by its evolutionary history.

To determine how the distribution and diversity of continuous traits are consistent with the phylogeny of *Trichoderma*, we calculated Blomberg's K [62] and Pagel's Lambda [63, 64], two different quantitative measures of the "phylogenetic signal". Using the "phylosig" function in "phytools" v1.3–5 [184] in we determined both the Blomberg's K (method = 'K') and Pagel's Lambda (method = 'lambda') for continuous traits such as total number of gene clusters, total number of separate gene cluster classes, total chitinase genes, total number of separate chitinase gene classes, and total number of mycoparasitism genes. Similar to Fritz and Purvis' D statistic, these analyses also incorporate the branch length and structure of a phylogenetic tree to determine the measure of how much the evolutionary history of taxa history drive the distribution and pattern of a particular trait.

Blomberg's K is a scaled ratio of (variance among the trait counts between the different species/contrasts variance); when phylogenetic signal is strong, the contrasts variance is low. When Blomberg's K is equal to 1, this indicates that the model is completely equal to Brownian motion, or that the evolutionary history completely explains the distribution and pattern of a particular trait. When K < 1, there is less strength of phylogenetic signal than expected under Brownian model evolution. When K > 1, there is a greater signal than expected. Pagel's Lambda is a parameter that scales between 0 and 1, indicating the correlations between species relative to the relationship expected when the trait is distributed according to the Brownian motion model of evolution. When Pagel's Lambda is equal to 1, this indicates that the model is completely equal to Brownian motion. When Pagel's Lambda is equal to 0, this indicates that there is no phylogenetic influence on the distribution and pattern of a particular trait. As

Pagel's Lambda decreases from 1 to 0, this is indicating a progressively lesser influence of the "phylogenetic signal" on the count of a particular trait. Pagel's Lambda and Blomberg's K test for the null hypothesis that there is no phylogenetic signal, therefore a significant p-value indicates that there is significant phylogenetic signal for a particular genomic feature. We used an arbitrary cutoff value of 0.8 for Blomberg's K and Pagel's Lambda; values equal to or lower than 0.8 are considered to have the potential of indicating gene clusters with selection in their evolutionary history. A significant p-value indicates that a particular genomic feature's distribution is consistent with its evolutionary history (follows Brownian motion model evolution).

We determined the ecological distribution bias of each trait by calculating the ratios for the mycotroph:saprotroph genomes (average count of each trait in mycotroph genomes/average count of each trait in saprotroph genomes) and endophyte:non-endophyte genomes (average count of each trait in endophyte genomes/average count of each trait in saprotroph genomes). We considered a comparative 2-fold increase (ratio value > 2) in the endophyte:non-endophyte (E:NE) and mycotroph:saprotroph (M:S) ratio to indicate genomic traits that were significantly enhanced in a particular *Trichoderma* ecology. The resulting ratios and respective phylogenetic signal statistic for each genomic feature was plotted on a scatter plot. Genomic features which had a significance level indicating that their distributions were consistent with *Trichoderma* phylogeny (consistent with Brownian motion evolution, or neutral evolution) were colored grey, points which were not entirely consistent with *Trichoderma* phylogeny and therefore were of potential interest for underpinning a particular trait were colored black.

## Ergot alkaloid BGC phylogenetic and synteny analyses

To study the evolution of the Brevicompactum clade ergot alkaloid BGC, we implemented the Cluster Reconstruction and Phylogenetic Analysis Pipeline (CRAP) across a database of 2,297 GenBank and MycoCosm [185] published fungal genomes using Mycotools v0.27.15 (gitlab. com/xonq/mycotools). We obtained the complete Brevicompactum clades' clusters by synthesizing non-overlapping gene coordinates from our annotations with MycoCosm's (Portal Tribre1, Triaru1). *easE* and *easF* were originally fused in Tribre1's annotation, so to construct robust phylogenies of each gene, we split the gene via exonerate [186] v2.2.0 (maxintron 100, percent 10, protein2genome:bestfit model) referencing *X. nigripes easE* and *easF*.

We queried the refined *T. brevicompactum* cluster using CRAP, which first obtained homologous genes via BLASTp [187] with a minimum bitscore threshold 30. To increase phylogenetic reconstruction throughput, we set CRAP to implement sequence similarity clustering to truncate groups of greater than 250 homologous sequences. The sequence clustering module iteratively implements *mmseqs2* v14.7e284 [188] *easy-cluster* with a minimum query coverage +/- 40% and adjusts the minimum sequence identity of each iteration until a group of fewer than 250 sequences was retrieved. Each query's truncated homolog group was subsequently aligned via mafft v7.487 (Katoh and Standley 2013) with default settings, trimmed using ClipKIT [189] v.1.3.0 with default settings, and initial phylogenies constructed via FastTree [180] v2.1.10 with default settings. We used the full 379 homologs retrieved by BLAST for Tribre1_123779 because the sequence similarity clusters were too granular (12 genes) to contextualize its evolution. FastTree phylogenies were rooted upon an outgroup branch identified by a larger module of homologs detected via *mmseqs easy-cluster*. Alternatively, these phylogenies were midpoint rooted if the initial BLASTp retrieved homologs were under 250 sequences.

To examine shared synteny with the Brevicompactum clade ergot alkaloid BGCs, CRAP extracted 50 kB up-/downstream of each homolog, truncated the loci boundaries to homologs of query genes, and mapped the resulting synteny diagrams on each query phylogeny. To

generate robust IQ-TREE phylogenies, we extracted highly supported clades ($> 0.98$ Shimo-daira-Hasegawa test) from FastTree phylogenies that contained known ergot alkaloid BGC homologs. Alternatively, all sequences from the FastTree phylogeny were used in IQ-TREE when the FastTree was rooted with known ergot alkaloid BGC homologs [139]. We inferred robust maximum likelihood phylogenies via implementing IQ-TREE v2.0.3 with 1000 ultrafast bootstrap iterations after aligning and trimming as described above.

To investigate the origin of *Trichoderma* ergot alkaloid BGCs, we compared the topology of the IQ-TREE gene phylogenies with a species phylogeny of genomes that contain chanoclavine cyclase oxidoreductase positional homologs. We constructed the species phylogeny via IQ-TREE using a multigene partition phylogeny of 13 profile hidden markov protein models [190] that were recovered from 2,296 / 2,297 fungal genomes. We obtained the top hit of these protein models from each genome as described in [191] using *db2search.py* with an e-value cutoff of 0.01 and minimum query coverage of 50%. In preparation for phylogenetic recon-struction, the retrieved proteins were aligned via mafft, trimmed via ClipKIT (—gappy 0.7), and submitted to ModelFinder [192] via *fa2tree.py* (gitlab.com/xonq/mycotools). The best model for each protein was used to build the multigene partition phylogeny from the concatenated trimmed alignments. To visualize shared synteny with respect to the species' phylogeny, we submitted loci obtained from the *easA* homologs to clinker v0.0.24 [193] with minimum alignment identity set to 50%. We compared the log-likelihood values of con-strained *easA* phylogenies that forced monophyly of either Xylariales, Hypocreales, or Euro-tiales to the optimal tree using an approximately unbiased test [71] implemented in IQ-TREE (S7 Table).

The genome assembly, annotations, and other resource-intensive analyses were performed using the Ohio Supercomputer Center [194] services.

## Supporting information

**S1 Fig. The dataset of 39 Trichoderma genomes contains 5 clades with different combina-tions of lifestyles and nutritional modes.** This consensus phylogenomic tree was created from 154 SCO IQ-TREEs which had >98% BP, using 10,000 bootstraps. Values at nodes repre-sent the IQ-TREE maximum likelihood BP, Internode Certainty (IC), and Internode Certainty All (ICA) (BP/IC/ICA). Internode Certainty (IC) and Internode Certainty All (ICA) values were obtained from the corresponding RAxML majority rule extended consensus tree. Species lifestyle is indicated by presence/absence of color-coordinated cell, nutritional mode is indi-cated by either a circle (saprotroph) or a star (mycotroph). Endophyte cells half-filled indicated a potential endophytic lifestyle for that species. Branch lengths are not proportional to sequence divergence.
(JPG)

**S2 Fig. Genome quality was not correlated with number of metabolic gene clusters identi-fied.** Genome quality is not correlated with the number of (A) DGC families (Pearson's, $p > 0.05$) (B) BGC families (Pearson's, $p > 0.05$), and (C) number of metabolic genes (Pear-son's, $p > 0.05$) identified in the *Trichoderma* genomes. Each point represents a *Trichoderma* isolate. Point fill indicates nutritional mode, point shape indicates the clade to which the isolate belongs, and color indicates whether the *Trichoderma* species is recorded as having an endo-phytic lifestyle. Gray shaded areas indicate the 95% confidence interval.
(PDF)

**S3 Fig. Genomes with greater numbers of DGCs also contained greater numbers of BGCs.**
(A) The number of DGC families identified in each genome positively correlates with the

number of identified biosynthetic gene cluster families (Pearson's, p < 0.005). (B) The proportion of DGC families to total number genes per genome positively correlates with the proportion of DGC families to total number genes per genome (Pearson's, p < 0.005). Each point represents a *Trichoderma* isolate. Point fill indicates nutritional mode, point shape indicates which clade to which the isolate belongs, and color indicates whether or not the isolate has a possible endophytic lifestyle. Gray shaded areas indicate the 95% confidence interval.
(PDF)

**S4 Fig. Larger genomes contain more gene clusters and metabolic genes.** There is a significant correlation between genome length (bp) of the 38 *Trichoderma* genomes and the diversity of (A) DGC families (Pearson's, p ≤ 0.05), (B) BGC families (Pearson's, p ≤ 0.05), and (C) proportion of metabolic genes to total genes (Pearson's, p ≤ 0.05). Points highlighted in red are isolates identified to have an endophytic lifestyle, points in black did not have an identified endophytic lifestyle, and points in blue were species with a potential endophytic lifestyle. Mycotrophic genomes are represented by empty points, saprotrophic genomes are represented by filled points. Gray shaded areas indicate the 95% confidence interval.
(PDF)

**S5 Fig. Larger genomes contain a greater proportion of gene clusters to genes.** There is a significant correlation between genome length (bp) of the 38 *Trichoderma* genomes and the diversity of (A) proportion of biosynthetic genes to BGC clusters (Pearson's, p ≤ 0.05) and (B) proportion of DGC families to total genes (Pearson's, p ≤ 0.05). Points highlighted in red are isolates identified to have an endophytic lifestyle, points in black did not have an identified endophytic lifestyle, and points in blue were species with a potential endophytic lifestyle. Mycotrophic genomes are represented by empty points, saprotrophic genomes are represented by filled points. Gray shaded areas indicate the 95% confidence interval.
(PDF)

**S6 Fig. Genomes with more genes contain greater numbers of metabolic gene clusters.** The overall number of genes identified in a *Trichoderma* genome is correlated with the number of (A) identified DGCs (Pearson's, p < 0.005) and (B) BGCs (Pearson's, p < 0.005). Gray shaded areas indicate the 95% confidence interval.
(PDF)

**S7 Fig. Endophyte genomes contain greater numbers of metabolic gene clusters and mycoparasitism genes.** The average number metabolic gene clusters is significantly different between endophytic and non-endophytic *Trichoderma* genomes for both (A) DGCs (Wilcoxon, p ≤ 0.05) and (B) BGCs (two-sample t-test, p ≤ 0.05). There is no significant differences between the number of (C) chitinase genes for endophytic and non-endophytic *Trichoderma* genomes (Wilcoxon, p > 0.05). The number of mycoparasitism genes is significantly higher in (D) endophyte genomes (Wilcoxon, p ≤ 0.05). (E) The number of metabolic genes per genome were higher in endophytes than in non-endophyte genomes (two-sample t-test, p ≤ 0.05). (F) The number of metabolic genes per BGC was not significantly different between endophyte and non-endophyte genomes (two-sample t-test, p > 0.05).
(PDF)

**S8 Fig. Endophytic *Trichoderma* genomes contain significantly greater numbers of ARD, NAD, and FAD DGC.** Differences between endophytic and non-endophytic DGC class counts were calculated using Wilcoxon rank sum test corrected with Holm-Bonferroni method. Endophytic genomes have significantly greater ARD, NAD, and FAD (each p ≤ 0.05), significance differences between groups are indicated with an asterisk. No

significant difference between endophyte and non-endophyte DGC count is indicated with
"NS" (p > 0.05).
(PDF)

**S9 Fig. Total metabolic cluster counts and individual metabolic cluster class distributions
are reflected by *Trichoderma* phylogeny.** Most evaluated DGC and BGC class distributions
were determined to be consistent with Brownian motion evolution as determined by their (A)
Blomberg's K value and significance (p-value ≤ 0.05) and (B) Pagel's Lambda value and significance (p ≤ 0.05). Black points in the shaded area of the graph would indicate DGC classes
which exhibit overdispersal in the *Trichoderma* phylogeny as well as overrepresentation in
endophytic genomes.
(PDF)

**S10 Fig. The distribution of individual DGC families is consistent with *Trichoderma* phylogeny.** Most DGC families did not have a significant result when analyzing each DGC's Fritz
and Purvis' D-statistic (p > 0.05), indicating that the distribution most DGC families followed
Brownian motion evolution. The DGC families most overrepresented in endophyte genomes
(E:NE > 3), families with a significant D-statistic (p ≤ 0.05), families with an very low D-statistic (D-estimate < -2) are labeled. Black points in the shaded area of the graph would indicate
DGC families which exhibit overdispersal in the *Trichoderma* phylogeny as well as overrepresentation in endophytic genomes.
(JPG)

**S11 Fig. Metabolic gene cluster content and proportions vary among *Trichoderma* clades.**
There are significant differences between the *Trichoderma* clades for total count of (A) DGCs
per genome (p ≤ 0.05) and (B) BGCs per genome (p ≤ 0.05). There are also significant differences between clades for the (C) proportion of DGCs to total gene number in each genome
(p ≤ 0.05) and (D) proportion of BGCs to total gene number in each genome (p ≤ 0.05). Significant differences were determined using Kruskal-Wallis test, post hoc differences were
determined with Dunn's test with Bonferroni correction.
(JPG)

**S12 Fig. The number and distribution of different BGC families varies across the Trichoderma phylogeny.** Heat map showing presence/absence of biosynthetic gene cluster families
for each *Trichoderma* genome. Singleton BGCs, as well as the highly fragmented *T. cf. atroviride* (LU140) genome, are included in this figure. The total number of BGC per genome, as well
as the distribution of fungal-RiPP, other hybrid, and beta-lactone clusters, are unchanged
from those in Fig 1 and are thus excluded for visualization purposes. Species lifestyle is indicated by presence/absence of color-coordinated cell, nutritional mode is indicated by either a
circle (saprotroph) or a star (mycotroph). Endophyte cells half-filled indicated a potential
endophytic lifestyle for that species.
(PDF)

**S13 Fig. Endophytic genomes are enriched in T1PKS, NRPS-like, betalactone, and fungal-RiPP BGCs.** Significant differences were determined by the Wilcoxon rank sum test corrected
with Holm-Bonferroni method. No significant differences between groups are indicated by
"NS". Outliers within groups are labeled.
(JPG)

**S14 Fig. The distribution of most individual BGC families is consistent with *Trichoderma*
phylogeny.** The majority of evaluated BGC families did not have a significant result when analyzing each's Fritz and Purvis' D-statistic (p > 0.05), which indicated Brownian motion

evolution. The BGC families that did not exhibit Brownian motion model evolution were not found enriched in endophyte genomes (E:NE < 2). The BGC families most overrepresented in endophyte (E:NE value > 6) and DGCs overdispersed in the *Trichoderma* phylogeny (D-statistic > 1) are labeled. Black points in the shaded area of the graph would indicate BGC families which exhibit overdispersal in the *Trichoderma* phylogeny as well as overrepresentation in endophytic genomes.
(JPG)

**S15 Fig. Genomes with greater numbers of genes and larger genomes contain more mycoparasitism-related genes.** The total number of mycoparasitism genes per genome is positively correlated with (A) total genes in a Trichoderma genome (Pearson's, p ≤ 0.05) and (B) genome length (Pearson's, p ≤ 0.05). (C) There is no relation to genome quality (N50) and number of identified mycoparasitism-related genes per genome (Pearson's, p > 0.05). Gray shaded areas indicate the 95% confidence interval.
(JPG)

**S16 Fig. Metabolic gene clusters and mycoparasitism genes are enriched in mycotrophic genomes.** The average number of metabolic gene clusters is significantly different between mycotrophic and saprotrophic Trichoderma genomes for both A) DGCs (2-sample t-test, p ≤ 0.0005) and B) BGCs (2-sample t-test, p ≤ 0.0005). The total number of C) chitinase genes is higher in mycotrophic genomes (2-sample t-test, p ≤ 0.0005), as are the D) total number of mycoparasitism genes (2-sample t-test, p ≤ 0.0005). Outliers are labeled and significant differences are indicated with asterisks.
(JPG)

**S17 Fig. Some mycoparasitism genes are overrepresented in mycotrophs and overdispersed in *Trichoderma*.** (A) The mycotroph to saprotroph gene ratio (M:S) and Blomberg's K value suggests that five mycoparasitism genes are overdispersed in *Trichoderma* (K < 0.8, p > 0.05) and enriched in genomes with a mycotrophic nutritional mode (M:S > 2). (B) The mycotroph to saprotroph gene ratio (M:S) and Blomberg's K value suggests that three mycoparasitism genes are overdispersed in *Trichoderma* (Lambda < 0.8, p > 0.05) and enriched in genomes with a mycotrophic nutritional mode (M:S > 2). Black points in the shaded area of the graph would indicate mycoparasite genes which exhibit overdispersal in the *Trichoderma* phylogeny as well as overrepresentation in mycotrophic genomes.
(JPG)

**S18 Fig. Amount and percent of mycoparasitism-related genes differ between *Trichoderma* clades.** (A) Longibrachiatum contains the fewest amount of mycoparasitism-related genes while Virens and section Trichoderma contain the highest amount (Kruskal-Wallis, correction with Bonferroni, p ≤ 0.05). (B) Harzianum, Longibrachiatum, and Virens contain the lowest percent of mycoparasitism genes to the total number of genes per genome, and Brevicompactum and section Trichoderma contains the highest (ANOVA, correction with Bonferroni, p ≤ 0.05).
(JPG)

**S19 Fig. Phylogenic correction of mycoparasitism data still indicate separation based on *Trichoderma* clade.** (A) Phylogenetic tree and relative signal of phylogenetically-corrected global principal component 1 (PC1) and global principal component 2 (PC2) for each *Trichoderma* genome. (B) Top 10 mycoparasitism genes contributing to PC1 signal and top 11 mycoparasitism genes contributing to PC2 signal. (C) Eigenvalues for the phylogenic correction

PCA (pPCA) global principal components (PCs).
(PDF)

**S20 Fig. *Trichoderma* genomes have similar mycoparasitism gene repertoires to other genomes within the same clade.** Principal components analysis (PCA) of the distribution of the 746 mycoparasitism-related genes; the inherent phylogenetic bias within the *Trichoderma* dataset was corrected with phylogenetic principal components analysis (pPCA). Plotting of the global principal components values for each genome shows that *Trichoderma* genomes are grouping based on *Trichoderma* clade rather than nutritional mode or lifestyle.
(JPG)

**S21 Fig. Larger genomes and genomes that contain more genes contain greater numbers of chitinase genes.** The overall number of genes identified in a *Trichoderma* genome is correlated with the number of A) identified chitinase genes (Pearson's, $p \leq 0.05$) and B) genome length (Pearson's, $p \leq 0.05$), but not with C) genome quality (Pearson's, $p \geq 0.05$). Gray shaded areas indicate the 95% confidence interval.
(JPG)

**S22 Fig. Endophytes only differ in chitinase BII gene/total gene proportion compared to non-endophytes.** (A) Endophyte genomes do not significantly differ in any chitinase class overall count when compared with non-endophyte genomes (Wilcoxon, $p > 0.05$) (B) (A) Endophyte genomes only have a significantly higher proportion of chitinase class BII genes to total genes per genome as compared to non-endophyte genomes (Wilcoxon, $p \leq 0.05$).
(PDF)

**S23 Fig. No chitinase classes are overrepresented in endophytic *Trichoderma* genomes.** Most evaluated chitinase class distributions were determined to be consistent with Brownian motion evolution as determined by their (A) Blomberg's K value and significance (p-value $\leq 0.05$) and (B) Pagel's Lambda value and significance (p $\leq 0.05$). Black points in the shaded area of the graph would indicate chitinase gene classes which exhibit overdispersal in the *Trichoderma* phylogeny as well as overrepresentation in endophytic genomes.
(JPG)

**S24 Fig. Mycotrophs differ in both count and proportion of chitinase classes compared to saprotrophs.** (A) Mycotroph genomes have significantly greater numbers of BI, BII, and CI class chitinase genes compared to saprotroph genomes (2-sample t-test, $p \leq 0.05$). (B) Saprotroph genomes have significantly greater percentage of total gene number for AII, AIV, AV, BI, BII, BV, CI, and CHITD class chitinase genes compared to mycotroph genomes (2-sample t-test, $p \leq 0.05$).
(JPG)

**S25 Fig. Most chitinase class distributions were consistent with *Trichoderma* phylogeny.** Most evaluated chitinase class distributions were determined to be consistent with Brownian motion evolution as determined by their (A) Blomberg's K value and significance (p-value $\leq 0.05$) and (B) Pagel's Lambda value and significance (p $\leq 0.05$). Black points in the shaded area of the graph would indicate chitinase gene classes which exhibit overdispersal in the *Trichoderma* phylogeny as well as overrepresentation in mycotroph genomes.
(JPG)

**S26 Fig. Genomes in different *Trichoderma* clades differ in their chitinase gene content and proportion.** There are significant differences between *Trichoderma* clades in their (A) number of chitinase genes (p $\leq 0.05$) and (B) percent of chitinase genes (p $\leq 0.05$). Outlier

genomes are labeled. Significant differences are indicated by different letters. Differences were calculated with ANOVA, post hoc tested with Tukey's HSD.
(JPG)

**S27 Fig. *Trichoderma* clades differ in total amount and proportion of LysM and CBD content.** (A) Virens has highest number of LysM and section Trichoderma has the fewest (ANOVA, corrected with Holm-Bonferroni, $p \leq 0.05$). (B) Virens has highest number of CBD; Brevicompactum and section Trichoderma have the fewest (ANOVA, corrected with Holm-Bonferroni, $p \leq 0.05$). (C) Section Trichoderma has the lowest percentage of LysM to total gene count (ANOVA, corrected with Holm-Bonferroni, $p \leq 0.05$). (D) Section Trichoderma has the lowest percentage of CBD to total gene count (ANOVA, corrected with Holm-Bonferroni, $p \leq 0.05$). Significant differences are indicated by different letters, as determined by Tukey's HSD.
(JPG)

**S28 Fig. Domain architecture of the 26 chitinases in *T. endophyticum* strain PP24.** Chitinase classes are indicated according to (Goughenour 2020). GH-18 domains are represented by black ovals, chitin-binding domains (CBD) are represented as green pentagons, Lysin domains (LysM) are represented as orange hexagons, and an RTA1 Superfamily domain is represented by a blue cylinder.
(PDF)

**S29 Fig. *Trichoderma* genomes with multiple potential lifestyles have more metabolic gene clusters.** Genomes of *Trichoderma* species recorded as having multiple lifestyles have (A) a significantly greater diversity of DGC (two-sample t-test, $p \leq 0.05$) and (B) a significantly greater number of BGC (two-sample t-test, $p \leq 0.05$) as compared to genomes of *Trichoderma* species recorded as only having a single potential lifestyle.
(JPG)

**S1 Table. Species name, strain identifier, source of genomic reads, lifestyle information, nutritional mode, and geographic origin of each *Trichoderma* isolate used in this study.**
(DOCX)

**S2 Table. Genome statistics of the 39 *Trichoderma* isolates.**
(DOCX)

**S3 Table. Mycoparasitism gene orthogroups of interest found in the highest endophyte: non-endophyte ratio.**
(DOCX)

**S4 Table. Mycoparasitism gene orthogroups of interest found in the highest mycotroph: saprotroph ratios as well as orthogroups only found in mycotrophs.**
(DOCX)

**S5 Table. Placement of Brevicompactum clade ergot alkaloid BGC gene phylogenies.**
(DOCX)

**S6 Table. Functional annotations of the top 21 mycoparasitism genes contributing to the global PC1 and PC2 in the pPCA analysis.**
(DOCX)

**S7 Table. EasA phylogenies for the consensus phylogeny and the forced monophyletic phylogenies for Xylariales, Hypocreales, and Eurotiales.**
(DOCX)

## Author Contributions

**Conceptualization:** Kelsey Scott, Emile Gluck-Thaler, Priscila Chaverri, Jason Slot.

**Data curation:** Kelsey Scott, Zachary Konkel, Emile Gluck-Thaler, Guillermo E. Valero David, Coralie Farinas Simmt, Django Grootmyers, Priscila Chaverri, Jason Slot.

**Formal analysis:** Kelsey Scott, Zachary Konkel, Emile Gluck-Thaler, Coralie Farinas Simmt, Django Grootmyers, Jason Slot.

**Funding acquisition:** Jason Slot.

**Investigation:** Kelsey Scott, Zachary Konkel, Emile Gluck-Thaler, Guillermo E. Valero David, Coralie Farinas Simmt, Django Grootmyers, Jason Slot.

**Methodology:** Kelsey Scott, Zachary Konkel, Emile Gluck-Thaler, Jason Slot.

**Project administration:** Kelsey Scott, Jason Slot.

**Resources:** Jason Slot.

**Software:** Zachary Konkel, Emile Gluck-Thaler, Jason Slot.

**Supervision:** Kelsey Scott, Emile Gluck-Thaler, Jason Slot.

**Validation:** Kelsey Scott, Zachary Konkel, Emile Gluck-Thaler, Jason Slot.

**Visualization:** Kelsey Scott, Zachary Konkel, Emile Gluck-Thaler, Jason Slot.

**Writing – original draft:** Kelsey Scott, Jason Slot.

**Writing – review & editing:** Kelsey Scott, Zachary Konkel, Emile Gluck-Thaler, Guillermo E. Valero David, Coralie Farinas Simmt, Django Grootmyers, Priscila Chaverri, Jason Slot.

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
