## [Decision Letter · Decision Letter 0]

6 Jun 2023

PONE-D-23-08909Endophyte genomes support greater metabolic gene cluster diversity compared with non-endophytes in *Trichoderma*PLOS ONE

Dear Dr. Scott,

Thank you for submitting your manuscript to PLOS ONE. After careful consideration, we feel that it has merit but does not fully meet PLOS ONE’s publication criteria as it currently stands. Therefore, we invite you to submit a revised version of the manuscript that addresses the points raised during the review process.

This was an interesting study that is likely to have far-reaching implications for understanding fungal endophytes going forward. I apologize for the delay in getting this back to you, I lost contact with one reviewer. However, I have read the manuscript thoroughly and I agree with Reviewer 1's assessment of the work. Please attempt to address all their comments as throughly as possible.

We look forward to receiving your revised manuscript.

Kind regards,

Richard A. Wilson

Academic Editor

PLOS ONE

Journal Requirements:

"This work was supported by a grant from the National Science Foundation (DEB-1638999) to JCS and by a Marie Skłodowska-Curie Individual Fellowship (SEP-210615880-TRADEOFF) to EGT."

"This work was supported by a grant from the National Science Foundation (https://www.nsf.gov/) (DEB-1638999) to JCS and by a Marie Skłodowska-Curie Individual Fellowship (https://marie-sklodowska-curie-actions.ec.europa.eu/ ) (SEP-210615880-TRADEOFF) to EGT.

Reviewers' comments:

Reviewer's Responses to Questions

**Comments to the Author**

1. Is the manuscript technically sound, and do the data support the conclusions?

Reviewer #1: Yes

2. Has the statistical analysis been performed appropriately and rigorously? 

Reviewer #1: Yes

3. Have the authors made all data underlying the findings in their manuscript fully available?

Reviewer #1: Yes

4. Is the manuscript presented in an intelligible fashion and written in standard English?

Reviewer #1: Yes

5. Review Comments to the Author

Reviewer #1: The manuscript by Scott et al is a comprehensive comparative genomic analysis of Trichoderma genomes. I have a few suggestions for improvement.

1. Please define the terms mycotrophy and saprotrophy. For mycotrophy in particular, it was unclear to me how this term relates to endotrophy and mycoparasitism.

2. I'd encourage the authors to read through the results section again carefully and add additional references throughout. Example statements that are missing attribution include lines 217, 221, 244, 884. References for Bloomberg's K, Pagel's Lambda, Fritz and Purvis' D-statistic are provided in the methods, but i think it would be helpful to include their references in the results section as well (lines 280, 293).

3. Can the author's perform a topology test to confirm that a constrained phylogeny in which Trichoderma easA genes group with other Hypocreales is significantly worse than the unconstrained best tree? I think this would be a nice additional bit of evidence to support the claim of HGT.

6. PLOS authors have the option to publish the peer review history of their article (what does this mean?). If published, this will include your full peer review and any attached files.

Reviewer #1: No

---

## [Author Response · Author response to Decision Letter 0]

13 Jul 2023

Reviewer 1’s comments:

Please define the terms mycotrophy and saprotrophy. For mycotrophy in particular, it was unclear to me how this term relates to endotrophy and mycoparasitism.

Information has been added to the text as suggested (Lines 69-74). The introductory paragraph has been reworded to clarify the differences between lifestyle (endophyte/soil/wood/fungi) and nutritional mode (mycotroph aka mycoparasitism/saprotroph).

I'd encourage the authors to read through the results section again carefully and add additional references throughout. Example statements that are missing attribution include lines 217, 221, 244, 884. References for Bloomberg's K, Pagel's Lambda, Fritz and Purvis' D-statistic are provided in the methods, but i think it would be helpful to include their references in the results section as well (lines 280, 293).

References and comments have been added to the suggested lines.

Can the author's perform a topology test to confirm that a constrained phylogeny in which Trichoderma easA genes group with other Hypocreales is significantly worse than the unconstrained best tree? I think this would be a nice additional bit of evidence to support the claim of HGT.

Thank you for the recommendation to perform topology testing as we agree this would help support the inference of HGT. We conducted constraint analysis using the approximately unbiased topology tests (Table S7, referenced in revised text) and these indeed show topologies constrained to conform to vertical inheritance are significantly worse than the consensus tree. 

Additional comments regarding Editor’s edits/comments:

As requested, permit information added to methods (Lines 901-904). 

As requested, we would like to amend our Funding Statement as follows:

“This work was supported by grants from the National Science Foundation (https://www.nsf.gov/): (DEB-1638999/DEB-1638976) to JCS and PC and (DEB-925672/DEB-1019972) to PC. This work was also supported by a Marie Skłodowska-Curie Individual Fellowship (https://marie-sklodowska-curie-actions.ec.europa.eu/ ) (SEP-210615880-TRADEOFF) to EGT.

Abstract was amended so that the online submission form abstract and abstract in manuscript are identical.

The order of the reference list was changed due to the added requested citations, as well as the modified order of citations in the manuscript. 

Additional formatting changes (header font size, case) were made to the manuscript to comply with the PLOS ONE format guidelines.

---

## [Editor Report · Decision Letter 1]

17 Jul 2023

Endophyte genomes support greater metabolic gene cluster diversity compared with non-endophytes in *Trichoderma*

PONE-D-23-08909R1

Dear Dr. Scott,

We’re pleased to inform you that your manuscript has been judged scientifically suitable for publication and will be formally accepted for publication once it meets all outstanding technical requirements.

Kind regards,

Richard A Wilson

Academic Editor

PLOS ONE
---

## [Editor Report · Acceptance letter]

21 Jul 2023

PONE-D-23-08909R1 

Endophyte genomes support greater metabolic gene cluster diversity compared with non-endophytes in *Trichoderma*

Dear Dr. Scott:

I'm pleased to inform you that your manuscript has been deemed suitable for publication in PLOS ONE. Congratulations! Your manuscript is now with our production department. 

Kind regards, 

on behalf of

Dr. Richard A Wilson 

Academic Editor

PLOS ONE